

# Effects of land use and water quality on greenhouse gas emissions from an urban river system

Long Ho[1]*, Ruben Jerves-Cobo[1, 2, 3], Matti Barthel[4], Johan Six[4], Samuel Bode[5], Pascal Boeckx[5], Peter Goethals[1]

[1] Department of Animal Sciences, Ghent University, Gent, Belgium;
[2] PROMAS, Universidad de Cuenca, Cuenca, Ecuador;
[3] BIOMATH, Department of Data Analysis and Mathematical Modelling, Ghent University, Gent, Belgium
[4] Department of Environmental System`s Science, ETH Zurich, Zurich, Switzerland;
[5] Department of Green Chemistry and Technology, ISOFYS Group, Ghent University, Gent, Belgium

*Correspondence to*: Long Ho (Long.TuanHo@UGent.be)

**Abstract.** Rivers act as a natural source of greenhouse gases (GHGs) that can be released from the metabolisms of aquatic organisms. Anthropogenic activities can largely alter the chemical composition and microbial communities of rivers, consequently affecting their GHG emissions. To investigate these impacts, we assessed the emissions of $CO_2$, $CH_4$, and $N_2O$ from Cuenca urban river system (Ecuador). High variation of the emissions was found among river tributaries that mainly depended on water quality and neighboring landscapes. By using Prati and Oregon Indexes, a clear pattern was observed between water quality and GHG emissions in which the more polluted the sites were, the higher were their emissions. When river water quality deteriorated from acceptable to very heavily polluted, their global warming potential (GWP) increased by ten times. Compared to the average estimated emissions from global streams, rivers with polluted water released almost double the estimated GWP while the proportion increased to ten times for very heavily polluted rivers. Conversely, the GWP of good-water-quality rivers was half of the estimated GWP. Furthermore, surrounding land-use types, i.e. urban, roads, and agriculture, significantly affected the river emissions. The GWP of the sites close to urban areas was four time higher than the GWP of the nature sites while this proportion for the sites close to roads or agricultural areas was triple and double, respectively. Lastly, by applying random forests, we identified dissolved oxygen, ammonium, and flow characteristics as the main important factors to the emissions. Conversely, low impact of organic matter and nitrate concentration suggested a higher role of nitrification than denitrification in producing $N_2O$. These results highlighted the impacts of land-use types on the river emissions via water contamination by sewage discharges and surface runoff. Hence, to estimate of the emissions from global streams, both their quantity and water quality should be included.

## 1 Introduction

Via the biogeochemical cycles of carbon (C), nitrogen (N) and water, (in)organic carbon and nitrogen compounds are added from terrestrial biosphere to inland water bodies (Meybeck, 1982;Schimel, 1995). These compounds can be transformed into greenhouse gases (GHGs), such as $CO_2$, $CH_4$, and $N_2O$, by microbial degradation and metabolisms, making rivers an active





source of GHGs to the atmosphere (Butman and Raymond, 2011;Raymond et al., 2013;Ho and Goethals, 2020a). Particularly, $CO_2$ and $CH_4$ are released mainly via the decay of organic matter during bacterial decomposition processes while nitrifying and denitrifying microorganisms are considered major generators of $N_2O$ in inland water bodies (Daelman et

al., 2013). Besides acting as a natural source of GHGs, rivers also serve as conduits for the GHGs released from groundwater and sediments to the atmosphere (Hotchkiss et al., 2015). In total, it was estimated from global streams and rivers that their $CO_2$ emissions were $1.8 \pm 0.25$ Pg C yr$^{-1}$ (Raymond et al., 2013) while the size of inland water $CH_4$ and $N_2O$ evasions were 26.8 Tg C yr$^{-1}$ and 1.26 Tg N yr$^{-1}$, respectively (Kroeze et al., 2005;Beaulieu et al., 2011;Stanley et al., 2016).

Besides the natural inputs from terrestrial ecosystems, anthropogenic activities such as fertilization or wastewater discharges
can lead to elevated nutrient inputs which in turn can lead to an increase in GHG emissions from inland water bodies. In urban areas, land-use changes and the discharges from sewers and wastewater treatment plants (WWTPs) have deteriorated river water quality by causing extensive modification in biochemical reactions and hydro- and morphology characteristics (Damanik-Ambarita et al., 2018). These anthropogenic sources were estimated to account for at least 10% of the global $N_2O$ emissions from rivers to the atmosphere (Beaulieu et al., 2011). While the concern about environmental impacts and human
health from the discharges has extensively been investigated, very little attention has been paid for their impacts on GHG emissions. National standards of the effluent discharge of WWTPs have been set to protect human health and the environment; however, their impacts on receiving rivers with respect to the GHG emissions have been absent.

Although the acknowledgment of the GHG emissions from rivers was given earlier (Meybeck, 1982;Kling et al., 1992), the progress of its mechanistic understanding is still facing many challenges (Goldenfum, 2012). The challenges derive from the
complex biological processes in the water column of rivers, their intricate interactions with terrestrial ecosystems and various human activities along the rivers. Due to the current limited understanding, a strong spatial variation of GHG emissions was frequently found in rivers without a clear explanation (Musenze et al., 2014). Recently, the variation of GHG emissions was referred to as a function of river sizes and their connectivity with terrestrial ecosystems (Hotchkiss et al., 2015;Raymond et al., 2013;Rosamond et al., 2012). Other studies indicated that agricultural run-offs have increased the
GHG emissions from rivers (Smith et al., 2017), while recent findings showed that urban infrastructure may contribute to the elevated GHG emissions from urban rivers (Kaushal et al., 2014;Gallo et al., 2014). However, it remains vague how these different landscapes affect the GHG emissions from the connected rivers and different water qualities of the rivers can impact their contribution to climate change. From this perspective, this study aims to clarify the link between neighboring land-use types, water quality, and the GHG emissions of river systems. To this end, we conducted a sampling campaign at
the five tributaries of Cuenca river urban system, collecting information about not merely the concentrations of the three main GHGs, i.e. $CO_2$, $CH_4$, and $N_2O$, but also physiochemical, hydromorphological, and meteorological variables. Subsequently, at each sampling site, we calculated Prati and Oregon water quality indexes and categorized different types of adjacent landscapes to investigate the impacts of these factors on the variation of the GHG emissions. Thereby, the study was able to calculate how the contribution of the rivers to climate change changed over different water quality categories and



land use types. Furthermore, statistical analysis and random forests were applied to investigate the spatiotemporal variation of the GHG emissions and identify the main important factors of the variation.

## 2. Materials and Methods

### 2.1 Study area

The study area is located at the Cuenca River basin situated in the southern province of Azuay in the Andes of Ecuador. The
basin is composed of five main tributaries, i.e. Cuenca, Tarqui, Yanuncay, Tomebamba, and Machangara Rivers. The study area is 223 km², representing 13% of the Cuenca River basin. The city of Cuenca has a population of approx. 401.000 inhabitants in 2019. Two natural reserves are also located upstream from the Cuenca River basin: Cajas National Park and the Machangara-Tomebamba protected forest. Both are water sources for the Tomebamba, Yanuncay and Machangara Rivers (Jerves-Cobo et al., 2018b). The mean altitude of the study area is 2655 m a.s.l. The annual average air temperature is
16.3 °C and the average rainfall is about 879 mm per year (Jerves-Cobo et al., 2018a). The rainy season starts from the middle of February until the beginning of July and from the second half of September until the first two weeks of November, while the rest of the year constitutes the dry season (Jerves-Cobo et al., 2020b). The area of Cuenca, Machangara, Tarqui, Tomebamba, and Yanuncay is 95.92, 111.19, 138.98, 113.03, 113.81 km², respectively.

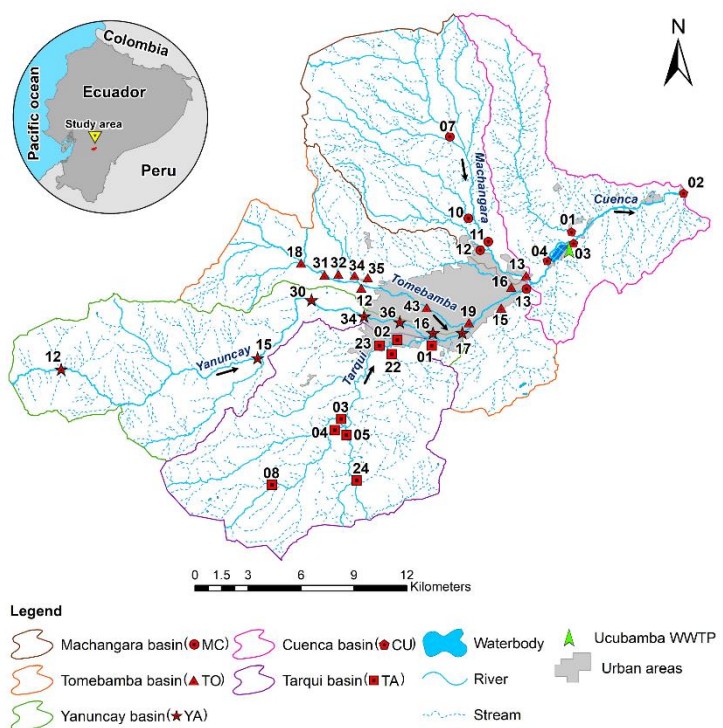

**Figure 1. Location of the study area in Ecuador and 36 sampling sites at the Cuenca urban river system.**



## 2.2 Field measurements

A sampling campaign was conducted from 17/09/2018 to 21/09/2018. During this period, samples were collected from 9.00 to 18.00. This course of time covers the whole period of daylight in Cuenca, ensuring the investigation of temporal effects on oxygen variation, hence, on the GHG emissions. 36 sites were sampled in the Cuenca river basin, splitting into the five

basins covering the whole urban river area as well as the river sources. Besides assessing the emissions of $CO_2$, $CH_4$, and $N_2O$, we also gathered physiochemical, hydro-morphological, and meteorological data. Specifically, water temperature, pH, dissolved oxygen (DO), turbidity, total dissolved solid (TDS), and chlorophyll *a* were determined by a handheld multiprobe (Aquaread-AP5000 version 4.07). Calibration was performed prior to sampling and supplemented with a regular check after sampling.

Water samples from all sampling sites were collected and stored in cool and dark containers and then preserved in a refrigerator before being analyzed for other variables in the Water and Soil Quality Analysis Laboratory at Cuenca University. Particularly, ammonium ($NH_4^+$), nitrite ($NO_2^-$), nitrate ($NO_3^-$) and orthophosphate ($PO_4^{3-}$) were determined spectrophotometrically (low-range Hach test kits with Hach DR3900). Moreover, water samples were kept frozen until shipment to Belgium for further analyses, i.e. biochemical oxygen demand ($BOD_5$), chemical oxygen demand (COD), total

nitrogen (TN), and total phosphorus (TP). Details of the Hach kits can be found in the Supplementary Material S1. Hydro-morphological information of the sites and their surroundings were collected, including land use, macrophytes, riparian vegetation, channel types, flow types, and sediment, via a modified field protocol of Jerves-Cobo et al. (2018b). Note that land use types surrounding the sampling sites were assessed using the modified field protocol based on the Australian River Assessment System physical assessment protocol (Parsons et al., 2002) and the United Kingdom and the Isle of Man River

Habitat Survey (Raven et al., 1997). In total, 17 variables were measured following different categories (Supplementary Material S2). River depth and velocity were measured at three points at each sampling site, two close to the riverbanks and one in the middle of the river. Meteorological data, including air temperature, solar radiation, rainfall, and wind speed, were obtained from the meteorological station of the University of Cuenca (-2.9050372°, -79.0124267°), located 7.8 km away from the Ucubamba WWTP and 0.7 km away from the city center.

To facilitate data accessibility, analysis, and visualization, we developed an interactive application using R **Shiny** package (Chang et al., 2015). This application allows customization of the application's user interface to provide an elegant environment for displaying user-input controls and simulation outputs (Wojciechowski et al., 2015). In short, the outputs can be instantaneously updated with the inputs, hence, the users can access, analyze, and visualize the collected data in a quick, flexible, and informative way. Detailed values of all variables in the 36 sampling sites are available online at https://water-

research.shinyapps.io/GHG_Cuenca/.



### 2.3 Dissolved gas concentrations

Dissolved GHG concentrations ($C_{aq}$) were measured using the headspace equilibration technique. Before the field campaign, 12 mL vials with airtight septa (Exetainer®, Labco Ltd, High Wycombe, UK) were pre-conditioned with 50μL of 50% ZnCl before capping and flushing with high purity $N_2$ (Alphagaz 2, Carbagas, Gümlingen, Switzerland). At each sampling, 6 mL

of water was pushed into the vials using a syringe after carefully removing air bubbles from the sample creating a headspace pressure inside the vial of ca. 2 atm. The headspace was analyzed for concentrations of $CO_2$, $CH_4$, and $N_2O$ using gas chromatography (Bruker, GC-456, Scion Instruments, Livingston, UK) equipped with a thermal conductivity detector, flame ionization detector, and electron capture detector. The instrument was calibrated for each gas using several sets of standards within each measurement run. Dissolved gas concentrations (μmol L$^{-1}$) were calculated by applying Henry's law, taking into

account the vial volume and headspace.

$$C_{aq} = p_a \times k_h \tag{1}$$

where $k_h$ is Henry's constant adjusted for lab temperature (mol m$^{-3}$ Pa$^{-1}$) and $p_a$ is the partial pressure of the gas in the headspace (Pa$^{-1}$).

### 2.4 Flux calculations

The flux (mg m$^{-2}$ d$^{-1}$) from the river water to the atmosphere of the three gasses assessed was calculated according to the model on gas exchange between air and water of Liss and Slater (1974):

$$Flux = k_0 \times (C_{aq} - C_{eq}) = k_0 \times (C_{aq} - p_a \times k_h) \tag{2}$$

where $k_0$ is the gas exchange coefficient (cm h$^{-1}$); $C_{aq}$ is the dissolved gas concentration (μmol L$^{-1}$), and $C_{eq}$ is the aqueous gas concentration in equilibrium with the atmosphere (μmol L$^{-1}$); $p_a$ is the partial pressure above the surface water at

equilibrium with atmosphere (Pa$^{-1}$); $k_h$ is Henry's law constant corrected in a given temperature (mol m$^{-3}$ Pa$^{-1}$);

The gas exchange coefficient $k_0$ was calculated as follows.

$$k_0 = k_{600} \times (Sc/600)^{0.5} \tag{3}$$

where $k_{600}$ is the gas exchange coefficient that was normalized to a common Schmidt number ($Sc$) of 600 (cm h$^{-1}$). $Sc$ is the Schmidt number that was calculated from an empirical third-order polynomial of Wanninkhof (1992) for the *in situ* water

temperature ($t_w$) for different gases as follows.

$$Sc_{CO2} = 1911.1 - 118.11 \times t_w + 3.4527 \times t_w^2 - 0.04132 \times t_w^3 \tag{4}$$

$$Sc_{CH4} = 1897.8 - 114.28 \times t_w + 3.2902 \times t_w^2 - 0.039061 \times t_w^3 \tag{5}$$

$$Sc_{N2O} = 2301.1 - 151.1 \times t_w + 4.7364 \times t_w^2 - 0.059431 \times t_w^3 \tag{6}$$



To calculate the $k_{600}$, an empirical function of Cole and Caraco (1998) that has been widely used counting for both wind
speed and temperature, was applied.

$$k_{600} = 2.07 + 0.215 \times U_{10}^{1.7} \tag{7}$$

where $U_{10}$ is the wind speed at 10-m height (m s$^{-1}$). Wind speed at 10-m height was equal to 1.22 times of wind speed at 1 m
above the water surface (Raymond and Cole, 2001;Wang et al., 2017).

The $k_h$ can be calculated by van't Hoff (1884) equation applied to Henry's law constant:

$$k_h(T) = k_h^0 \times exp\left[\frac{-\Delta_{sol}H}{R}\left(\frac{1}{T} - \frac{1}{T^0}\right)\right] \tag{8}$$

where $k_h^0$ is Henry's law constant at the reference temperature $T^0$ = 298.15 K; $\Delta_{sol}H$ is the enthalpy of dissolution. The
values of $k_h^0$ and $\Delta_{sol}H/R$ for the three GHGs are averaged from the list of their empirical values from different studies,
which can be found in Sander (2015). Particularly, the values of $k_h^0$ and $\Delta_{sol}H/R$ are $3.4 \times 10^{-4}$ (mol m$^{-3}$ Pa$^{-1}$) and 2400 (K)
for $CO_2$, $1.4 \times 10^{-5}$ (mol m$^{-3}$ Pa$^{-1}$) and 1600 (K) for $CH_4$, and $2.4 \times 10^{-4}$ (mol m$^{-3}$ Pa$^{-1}$) and 2600 (K) for $N_2O$. Partial pressure
$p_a$ of each gas was determined by injecting 20mL of air samples near the water surface into pre-evacuated 12 ml vials
(Exetainer®, Labco Ltd, High Wycombe, UK) which were subsequently analyzed in the Department of Environmental
Systems Science, ETH Zurich, Zurich, Switzerland using gas chromatography. Note that the flux calculation excluded the
contribution of ebullition that can be an important pathway of $CH_4$ emissions from certain aquatic sediment, such as lakes
and hydropower reservoirs (Bastviken et al., 2011;Tuser et al., 2017). This assumption was based on the absence of sediment
layers in most of the measured sites given that thick sediment layers under a shallow water column are major contributors of
ebullition process due to lower hydrostatic pressure and wave-induced perturbations (Bastviken et al., 2004).

Furthermore, we calculated the total emissions of each tributary per year by multiplying its flux to its total watershed area.
We calculated the fraction of the total emissions of all fluxes per year by converting the fluxes to $CO_2$ equivalent using the
values from the Fifth Assessment Report by the Intergovernmental Panel on Climate Change (IPCC) (2015: mass flux of
$CH_4$ multiplied by 28 and of $N_2O$ by 265) to determine the 100-year time horizon global warming potentials (GWP) of the
three gases released from rivers through diffusion (IPCC, 2014). The calculated values for the GHG emissions were
represented with mean and standard errors of the mean as we focused on the uncertainty around the estimate of the mean
measurement (Altman and Bland, 2005). We compared the estimated GHG emissions and their GWP from the sites with
different water quality categories and land use types to the average estimated values from global streams and water bodies
from the previous studies. In particular, Holgerson and Raymond (2016) estimated the emissions of $CO_2$ and $CH_4$ from
global freshwater bodies in a function of surface area using the measurement of the gases from 427 inland waterbodies
ranging in surface area from 2.5m$^2$ to 674km$^2$. We calculated the average values of the estimated emissions which were
equal to 984.6±160.8 mg-C m$^{-2}$ d$^{-1}$ and 4.2±1.0 mg-C m$^{-2}$ d$^{-1}$ for $CO_2$ and $CH_4$ emissions, respectively. Beaulieu et al. (2011)
also accounted for the surface area of the global streams when estimating their $N_2O$ emissions. In particular, the average $N_2O$
emission of the global streams was estimated to equal to 37 µg-N m$^{-2}$ h$^{-1}$ or 0.89 mg-N m$^{-2}$ d$^{-1}$.



## 2.5 Water quality indexes

To investigate the effects of water quality on the GHG emissions from receiving water bodies, water quality indexes were calculated. By aggregating the measurements of multiple water quality parameters, water quality index as a single number can be used to assess the quality of a water resource for serving different purposes (Lumb et al., 2011). Prati and Oregon indexes were calculated and compared. Particularly, Prati index, developed by Prati et al. (1971), is often used to evaluate surface water quality with a consideration of numerous pollutants while Oregon Index was developed by Dunnette (1979) and then modified by Cude (2001) to express ambient water quality for general recreational use. In this study, we calculated the basic Prati index of each sampling site by accounting for DO saturation, COD, and $NH_4^+$ concentration, and a modified Oregon Index containing six variables, i.e. water temperature, DO, $BOD_5$, pH, the total concentration of $NH_4$ and $NO_3$, and TP concentration. Details of their calculation can be found in the Supplementary Material S3. According to the Prati index, water quality can be ranked as good quality, acceptable quality, polluted, heavily polluted, and very heavily polluted. Similarly, five water quality categories can be found according to Oregon index, i.e. excellent, good, fair, poor, very poor.

## 2.6 Spatiotemporal variation of the GHG emissions

To investigate the spatiotemporal variation of the GHG emissions, we applied a linear mixed model (LMM) in R (R Core Team, 2014) using the **lme4** package (Bates et al., 2015). Not only accounting for fixed effects as linear regression models, LMM includes random effects that can take into account the spatiotemporal autocorrelations of observations (Dormann et al., 2007). Specifically, different sampling days and different tributaries were included to respectively assess the temporal and spatial variations of the collected samples. To do so, a three-level hierarchical mixed model was created, in which the unit of analysis, GHG emissions (level 1), is nested within rivers (level 2), which is in turn nested within the sampling days (level 3). The GHG emissions were log10 transformed and standardized. A final check for normality was done by using Cleveland plots (Supplementary Material S4). Moreover, homogeneity was checked via the residuals of the fitted model (Supplementary Material S5) while the assumption of multicollinearity was omitted due to the absence of fixed parameters. The impacts of the spatiotemporal autocorrelation are represented by the mean of intraclass correlation coefficient (ICC) as a measure describing the homogeneity of the observed GHG emissions within given clusters, i.e. river and sampling day (West et al., 2014). The ICC is determined via the variance components in the mixed model. Particularly, the sampling-day-level ICC ($ICC_{day}$) was calculated by dividing the variance of the random sampling-day effects ($\sigma_{sd}^2$) by the total random variation, consisting of $\sigma_{sd}^2$, the variance of the random effects associated with rivers nested within sampling campaign ($\sigma_r^2$) and the variance of residual ($\sigma^2$):

$$ICC_{day} = \frac{\sigma_{sd}^2}{\sigma_{sd}^2 + \sigma_r^2 + \sigma^2} \tag{9}$$





The value of $ICC_{day}$ is high when the total random variation is dominated by $\sigma_{sd}^2$, meaning that the GHG emissions measured among different sampling days tend to vary widely while these values among different rivers within a sampling campaign are relatively homogenous (Ho et al., 2018a). Similarly, the $ICC_{river:day}$ was calculated as follows.

$$ICC_{river:day} = \frac{\sigma_{sd}^2 + \sigma_r^2}{\sigma_{sd}^2 + \sigma_r^2 + \sigma^2} \tag{10}$$

If the value of $ICC_{river:day}$ is higher than $ICC_{day}$, it means that there was a large variation in the GHG emissions within the
same river as $\sigma_r^2$ is high (Ho et al., 2018a).

**2.7 Random forests**

Random forests (RFs) were first offered by Ho (1995) and then improved by Breiman (2001) via using an ensemble of a large number of decision trees. Offering sufficient accuracy, simple implementation, and high robustness, RFs have been largely accepted in the machine learning community (Tyralis et al., 2019). RFs were implemented in R via the **ranger**
package (Wright and Ziegler, 2017). To optimize the model, we tuned two essentials hyperparameters, including the minimal size of a node (*min.node.size*) and the number of candidate variables considered at each split (*mtry*) while the number of trees (*num.trees*) as a not tunable parameter was set at 500 (Probst et al., 2019). To do so, the **mlr** package of Bischl et al. (2016) was applied in parallel on eight CPU cores. The tuned model with optimal hyperparameters was run to identify the importance of variables for the GHG emissions. Permutation accuracy importance was preferred over the conventional
variable importance since it can deal with the drawbacks of the latter, e.g. bias towards continuous variables compared to categorical variables, and dividing up importance when variables are highly correlated (Strobl et al., 2007). The method of Janitza et al. (2018) for calculating permutation accuracy importance was applied in the **ranger** package.

**3. Results and discussion**

**3.1 Spatiotemporal variation of the GHG emissions**

We monitored five different tributaries in the Cuenca urban river system, including Cuenca, Machangara, Tarqui, Tomebamba, and Yanuncay, all showing strong variation in terms of GHG emissions (Figure 2). Converting the emissions to $CO_2$ equivalent, it appeared that Tomebamba tributary was the largest GHG contributor, accounting for 59.6% of the total emissions of the three gases per year from the whole river basin. Tarqui tributary ranked in the second place, contributing 21.2% of the total emissions per year, following by Cuenca tributary with 10.9%. Machangara and Yanuncay generated in
total less than 8% of the total emissions. Among the tributaries, the GHG emissions varied differently from one tributary to another. While high variation could be found in the largest GHG contributors, i.e. Tomebamba, Tarqui and Cuenca tributaries, the GHG emissions from Machangara and Yanuncay remained stable. Also noteworthy is that the mean value of the samples collected from Tomebamba, Tarqui and Cuenca were much higher than the median value, indicating the
emissions from the tributaries were positively skewed. The skewness was caused by several extremely high emissions
released from the sampling sites located in the three tributaries.

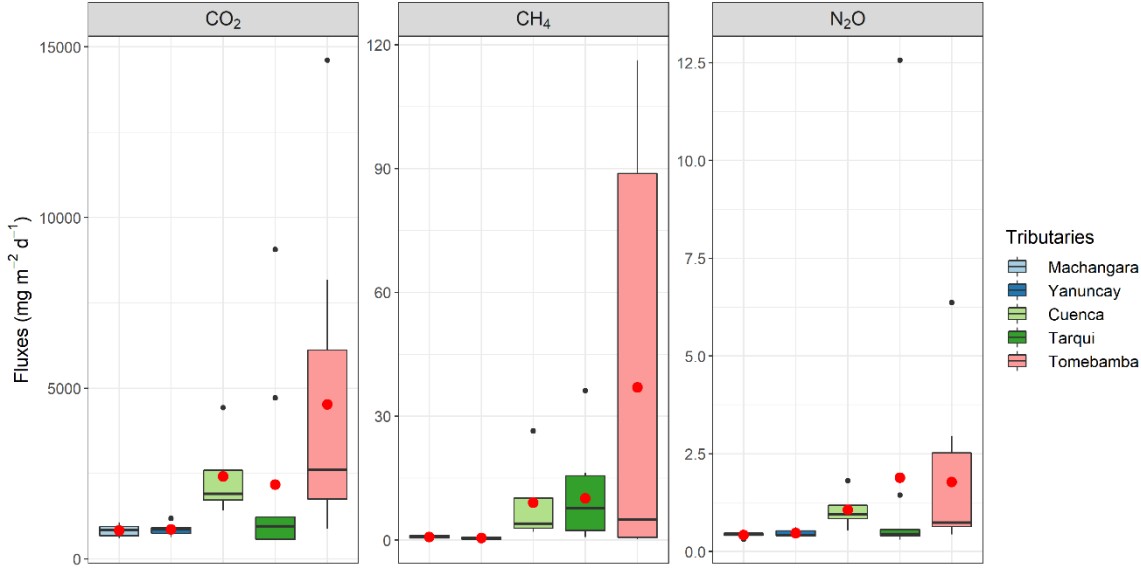

**Figure 2. Fluxes of the three greenhouse gases from the five tributaries of the Cuenca urban river system. Box plots display 10th, 25th, 50th, 75th and 90th percentiles, and individual data points outside the 10th and 90th percentiles. Red dots represent the arithmetic mean of the fluxes from different tributaries.**

High spatial variation of the GHG emissions was also indicated in the values of the obtained ICCs. Specifically, $ICC_{river:day}$

were 0.41 for $CO_2$, 0.47 for $CH_4$ and 0.24 for $N_2O$. These values were higher than the $ICC_{day}$, i.e. 0.19 for $CO_2$, 0.24 for $CH_4$

and 0.04 for $N_2O$. The differences between the two ICCs (around 20% of the total variation of the emissions) suggest a high

spatial variation of the emissions from the tributaries. Plus, the values of $ICC_{day}$ suggest a higher diurnal variation of the

$CO_2$ and $CH_4$ emissions compared to the stable $N_2O$ emissions across the sampling days since the variance of the random

diurnal effect explained only 4% of the total variation in the case of the $N_2O$ emissions. This contrast can be explained by

substantial temporal variation of DO level observed in Cuenca in the previous studies (Ho et al., 2018a;Ho et al., 2018b). In

particular, the production of $CO_2$ and $CH_4$ might depend stronger on the prevalence of DO as it controls the efficiency of

anaerobic/anoxic processes, which are mainly responsible for releasing $CH_4$, and highly correlated to the amount of $CO_2$

released from the algal metabolism (Ho et al., 2019). While $N_2O$ emissions, which are mainly from nitrification (Wunderlin,

2013), could remain stable in this study due to the high DO level in the tributaries.

**3.2 Effect of water quality on the GHG emissions**

Prati and Oregon Indexes were applied to assess the effects of water quality on the GHG emissions from the receiving water
bodies. According to the Prati Index, the rivers had higher water quality than the results obtained from the Oregon Index.



Particularly, 18 sampling sites were categorized in either good quality or acceptable quality following the Prati Index while only two sites were considered either good or fair water quality according to the Oregon Index. The reason for this difference is because of a heavy penalty for high organic matter and nutrient concentrations in the Oregon Index. Particularly, on average, the Oregon subindex values calculated for water temperature, DO, and pH, were relatively high, from fair to excellent water quality, which was in contrast to the low values of the Oregon subindex calculated for $BOD_5$, the total concentration of $NH_4$ and $NO_3$, and TP due to their high concentrations. These low subindex values made most of the sampling sites fall into the very poor category of water quality according to Oregon Index. Similarly, high concentrations of $NH_4$ were the main reason for polluted sites in the calculation of the Prati Index.

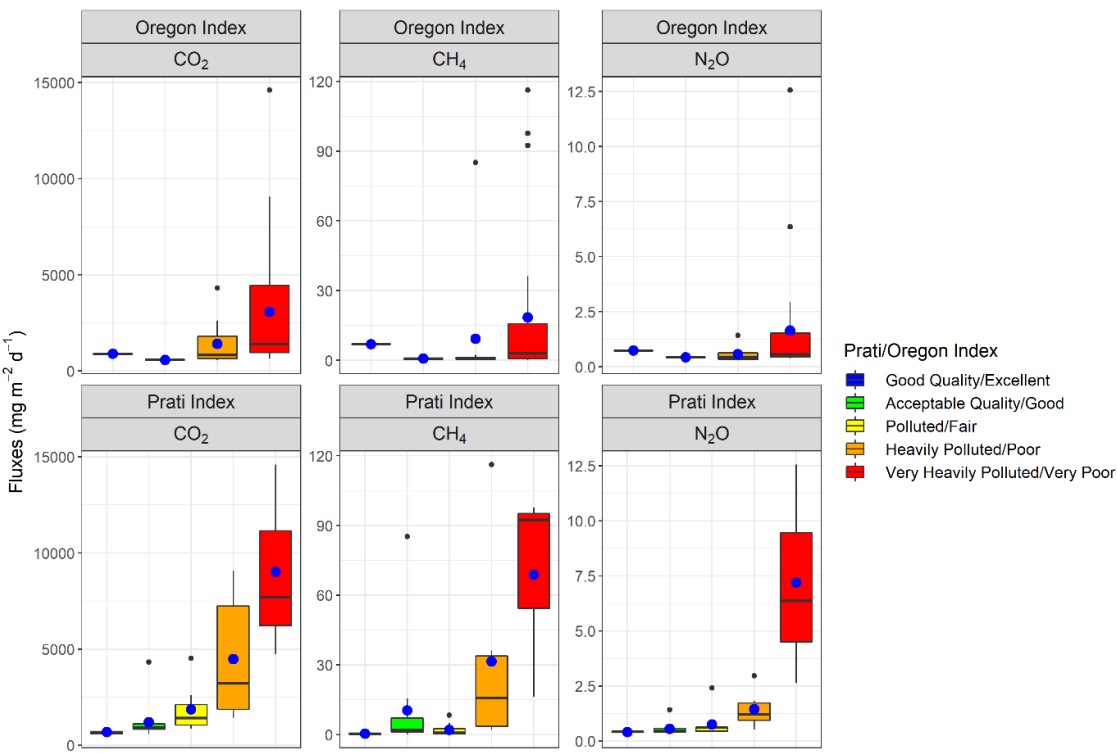

**Figure 3. Fluxes of the three greenhouse gases from the Cuenca urban river system in different water quality categories using Oregon and Prati Indexes. Box plots display 10th, 25th, 50th, 75th and 90th percentiles, and individual data points outside the 10th and 90th percentiles. Blue dots represent the mean of the fluxes in different water quality categories.**

Figure 3 shows the emissions of the three GHGs in different water quality categories using the Oregon and Prati Indexes. By comparing the mean of the emissions in the categories, a clear pattern between water quality and GHG emissions can be observed, in which the more polluted the sampling sites were, the higher were their GHG emissions. According to the Prati Index, when the water quality became worse by one level, the average of their $CO_2$ emissions was doubled up. In particular, the mean emissions from the sampling sites with good, acceptable, polluted, heavily polluted, and very heavily polluted



water quality were 673.4±46.3, 1203.7±290.7, 1865.3±390.3, 4483.1±1382.4, and 9014.8±2926.9 mg-C m$^{-2}$ d$^{-1}$, respectively. Similarly, when river water quality deteriorated from acceptable quality to very heavily polluted quality, the $CH_4$ emissions increased by up to seven times while the $N_2O$ emissions boosted by 13 times. As a result, the GWP of the very heavily polluted sites were almost ten times higher than that value of the sites with acceptable water quality, indicating the

considerably indirect negative impacts of polluted water bodies caused by anthropogenic activities. The GWP of the sites with different water quality based on Prati and Oregon Indexes can be found in Table 1. The emissions of contaminated sites were also much higher than the average estimated emissions of the global streams from the previous studies. It was estimated that the average $CO_2$ and $CH_4$ emissions of the global streams were 984.6±160.8 mg-C m$^{-2}$ d$^{-1}$ and 4.2±1.0 mg-C m$^{-2}$ d$^{-1}$, respectively (Holgerson and Raymond, 2016) while their $N_2O$ emissions were around 0.89 mg-N m$^{-2}$ d$^{-1}$ (Beaulieu et al.,

2011). Counting from these estimations, the average estimated GWP from the global inland waters is around 1337.3 ±189.1 mg $CO_2$ equivalent m$^{-2}$ d$^{-1}$. By comparison, rivers with polluted water quality could release almost double the average estimated GWP while if their water quality worsened to very heavily polluted, the proportion was up to ten times. On the other hand, when the rivers had a good water quality according to Prati Index, their GWP was only approximately half of the average estimated GWP while the GWP of acceptable-water-quality rivers was similar to the average estimated GWP.

Concerning the Oregon Index, apart from the abnormal high GHG emissions from one site with good water quality, it also appeared that when the more polluted sites were, the more GHGs could be produced. From fair to poor to very poor water quality, the $CO_2$ emissions increased from 562.9 to 1404.4 to 3071.9 mg-C m$^{-2}$ d$^{-1}$ while the $CH_4$ emissions increased from 0.7 to 9.2 to 18.4 mg-C m$^{-2}$ d$^{-1}$ and in case of the $N_2O$ emissions 0.4 to 0.6 to 1.6 mg-N m$^{-2}$ d$^{-1}$. This clear pattern suggests a new method for the global estimation of GHG emissions from water bodies accounting for both the quantity of the water bodies and their water quality. In this study, Prati Index appeared to be an optimal choice for indicating the impacts of water

quality on GHG emissions as illustrated in Table 1. Besides, as including only three variables, the application of Prati Index is more practical for the global estimation compared to Oregon Index.

**Table 1. Global Warming Potential (GWP) of the sites with different water quality based on Prati and Oregon Indexes.**

| Water Quality Categories (Prati/Oregon Index) | GWP of the sites-Prati Index (mg $CO_2$ equivalent m$^{-2}$ d$^{-1}$) | GWP of the sites-Oregon Index (mg $CO_2$ equivalent m$^{-2}$ d$^{-1}$) |
|---|---|---|
| Good Quality/Excellent | 792.5±55.2 | NA |
| Acceptable Quality/Good | 1441.9±507.6 | 695.2±NA |
| Polluted/Fair | 2124.1±472.5 | 1270.7±NA |
| Heavily Polluted/Poor | 5746.7±1977.3 | 1810.8±657.8 |
| Very Heavily Polluted/Very Poor | 12845.3±4430.9 | 4021.6±1057.5 |

Not available (NA) values were because there were no site with excellent water quality, one site of good water quality, and one site of fair

water quality according to the Oregon Index.





### 3.3 GHG emissions from different land-use types

Hydro-morphological variables, including land-use types around the rivers, bank erosion rate, flow variation, pool-riffle class, and shading level, were also monitored via a sampling protocol. The detailed distribution of the variables across the five rivers is shown in the mosaic plots S6.1-S6.5 in Supplementary Material S6. Due to the large sampling area, land-use 295 types widely varied while other variables remained relatively stable across the five tributaries. Particularly, urban and resident areas were dominant with around 55% of the total sampling areas, while forest and agriculture occupied 8-11% and 14-20%, respectively. Minor sampling area was surrounded by industrial factories and construction sites, with less than 5% each. Several riversides were next to the road, occupying 11-19% of the total sampling area. The 0distribution of the land-use types was not evenly among the rivers. Intensive urban activities can be found near to the Cuenca and from the middle to 300 the end of Tomebamba rivers. Conversely, Yanuncay and Machangara cross two natural reserves, i.e. Cajas National Park and the Machangara-Tomebamba protected forest, leading to their pristine water quality conditions natural (Jerves-Cobo et al., 2018b). Tarqui river locates near to agricultural irrigation and livestock production areas, causing their high nutrient and organic inputs (Jerves-Cobo et al., 2018b).

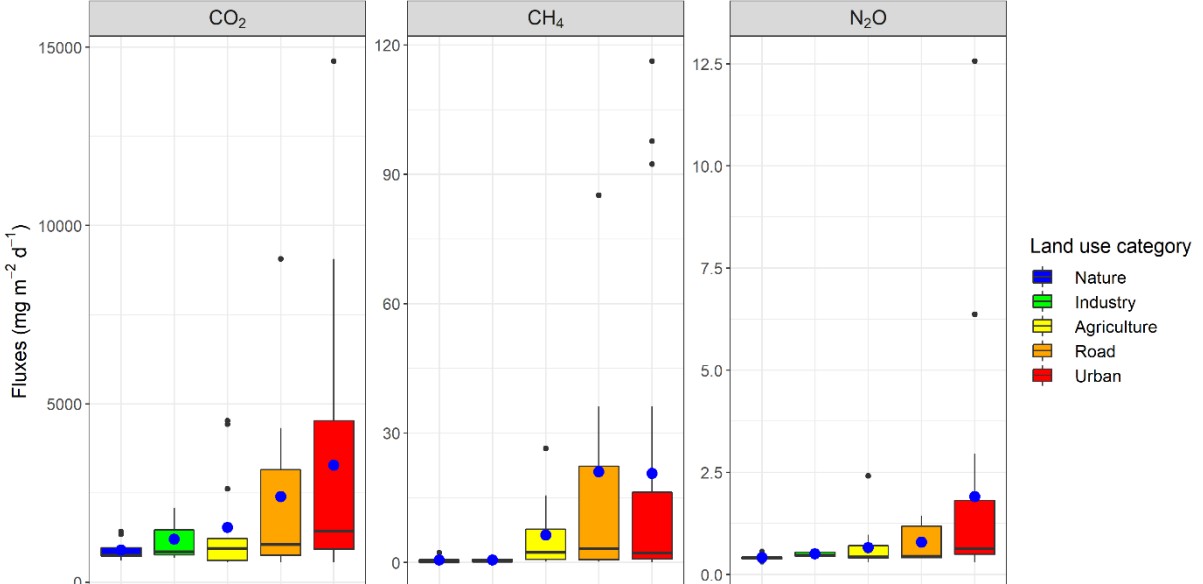

**Figure 4. Fluxes of the three greenhouse gases from the Cuenca urban river system in different land-use types. Box plots display 10th, 25th, 50th, 75th and 90th percentiles, and individual data points outside the 10th and 90th percentiles. Blue dots represent the arithmetic mean of the fluxes from different land use categories.**

Figure 4 shows an uneven distribution of the GHG emissions from the sampling sites close to different land-use types in the Cuenca urban river system. Looking at the mean of the emissions, it appeared that the sampling sites close to urban areas 310 released the most GHG emissions, i.e. $3276.5\pm611.2$ mg-C m$^{-2}$ d$^{-1}$ for $CO_2$, $20.6\pm6.1$ mg-C m$^{-2}$ d$^{-1}$ for $CH_4$, $1.9\pm0.5$ mg-N m$^{-}$





$^2$ d$^{-1}$ for N$_2$O, following by road, agriculture and industry areas. In contrast, nature areas appeared to affect the least the GHG emissions from the sites. Based on the emissions, the average GWP of the sites close to different land use types was calculated as shown in Table 2. The highest GHG productivity from the sites close to urban areas significantly boosted their GWP to 4356.4±912.5 mg CO$_2$ equivalent m$^{-2}$ d$^{-1}$ which was more than four time higher than the GWP of the sites close to

natural areas. Similarly, the GWP of the sites close to roads or agricultural areas was triple and double that value of the natural sites. In fact, the calculated GWP of the sites close to urban, road, and agricultural areas were much higher compared to the average estimated GWP of global streams while the sites close to nature areas showed a smaller GWP by 25%. These results highlight the indirect impacts of land-use change on increasing the GHG emissions from inland waters which are currently being omitted in land-use planning and resource management. The high emissions from sites close to urban, road,

and agricultural areas were highly likely to be caused by the pollutants from the point discharges of combined sewer overflows and Ucubamba WWTP in the Tomebamba and Cuenca tributaries as well as the surface runoff from roads and arable areas. For example, being the second-largest GHG contributor, Tarqui tributary had high nutrient and organic matter load as its surrounded landscapes mainly occupied by agricultural irrigation and livestock production (Jerves-Cobo et al., 2018b). Hence, it can be concluded that land-use types, i.e. urban, transportation systems, and agriculture, can have

considerable impacts on water quality and GHG emissions of the Cuenca river system. This conclusion is in line with previous studies on the role of the neighboring landscapes on GHG emissions from the rivers (Hotchkiss et al., 2015;Raymond et al., 2013;Rosamond et al., 2012). Smith et al. (2017) and, Yu and McCarl (2018) concluded that urban infrastructure greatly altered downstream water quality and was responsible for the variation of GHG emissions in rivers. Davidson et al. (2015) and Beaulieu et al. (2019) also reported that enhanced eutrophication in freshwater bodies will

increase their CH$_4$ emissions by 30-90% during the 21$^{st}$ century. Hence, it is important to consider the considerable, but often omitted, impact of human intervention on the GHG emissions from rivers via the alteration of nutrient loads and compositions.

**Table 2. Global Warming Potential (GWP) of the sites connected with different land use types**

| Land use types | GWP of the sites (mg CO$_2$ equivalent m$^{-2}$ d$^{-1}$) |
|---|---|
| Nature | 1024.2±121.8 |
| Industry | 1354.5±472.7 |
| Agriculture | 1890.1±495.6 |
| Road | 3197.1±1104.7 |
| Urban | 4356.4±912.5 |

**3.4 Main important variable on the variation of the GHG emissions**

In contrast to general machine learning tools focusing mostly on forecasting and prediction over-explanation, random forests allow for whitening the black-box with explicit interpretation of the obtained result through variable importance metrics





(Tyralis et al., 2019). In this case, permutation accuracy importance indicated the role of a variable in changing the model prediction of the GHG emissions. From Figure 5, DO appeared to be the most important factor in the production of GHGs in the Cuenca urban river basin. As mentioned previously, the DO level is a main controlling factor of anaerobic and anoxic

processes and highly correlated to algal metabolism, which can explain its first rank in the list of the variables. Moreover, $N_2O$ is mainly produced from nitrification and denitrification processes whose efficiency can strongly be affected by the availability of oxygen (Castro-Barros et al., 2017). Concerning the $N_2O$ pathways of ammonia-oxidizing bacteria, during nitrifier nitrification processes, high $O_2$ can enhance the contribution of hydroxylamine oxidation to $N_2O$ production while nitrifier denitrification is more active at oxygen limiting conditions (Schreiber et al., 2012). Moreover, a low amount of

oxygen causes the emissions of $N_2O$ during denitrification as $N_2O$ reductase is more sensitive to be inhibited by oxygen compared to nitrite reductase and nitric oxide reductase (Kampschreur et al., 2009;Schreiber et al., 2012). Additionally, an extra pathway of $N_2O$ emissions from the metabolism of green algae, which may be affected by nitrite concentration and photosynthesis repression, are not yet fully understood (Ho and Goethals, 2020b).

Nutrient concentrations also appeared as major important factors in GHG production in the rivers. The variation of $NH_4^+$

concentration, the input of nitrification process, affected the emissions of $N_2O$ the most while $NO_3^-$, the input of denitrification process, appeared to be a marginal controlling factor. IPCC assumed that nitrification produces $N_2O$ twice as much as denitrification in streams and rivers (Mosier et al., 1998). This assumption is highly likely true in this study as oxygen was relatively prevalent in the Cuenca urban river system, facilitating nitrification but inhibiting its consecutive steps in biological nitrogen removal processes. Beaulieu et al. (2011) and Rosamond et al. (2012) found no relationship between

$N_2O$ yield and stream water $NO_3^-$ and less than 1% of stream water nitrate subject to direct denitrification is converted to $N_2O$. Conversely, higher denitrification rates were found in the river floodplains as a result of nitrate-rich and oxygen-poor conditions in the flood water and flooded sediment (Venterink et al., 2003;Forshay and Stanley, 2005), which, however, was not the case in this study. Equally important, $NH_4^+$, the main nutrient input for algae, mosses, and macrophytes in the rivers, plays an important role in the variation of $CO_2$ emissions from rivers as plant nutrition partly determines the ratio of

photosynthesis to respiration. Theoretically, a high amount of ammonium can be an inhibitor for anaerobic processes but only in very high concentrations from 4.0 to 5.7 g $NH_3$-N $L^{-1}$ (Chen et al., 2008), which is not the case in this study. Despite being on the list of the most important factors on the emissions of all the three gases, $NO_2^-$ presence in the environment is often unstable and in a very low concentration, e.g. smaller than 0.01 mg $L^{-1}$ in this study. This condition is attributed to the fact that $NO_2^-$ is an intermediate oxidation state of nitrogen in the oxidation of ammonia to nitrate (nitrification), and in the

reduction of nitrate process (denitrification) (Hu et al., 2016). Also noteworthy is that COD appeared to have a relatively marginal impact on the variation of the three GHGs. The correlation coefficients between COD and the emissions of $CH_4$ and $CO_2$ were weak, i.e. 0.53 and 0.44, which is not in line with the conclusion on significant correlations between them in the study of Yang et al. (2015). Other factors indicating flow characteristics, such as turbidity, average velocity, average depth, and water temperature also affected the variation of the GHG emissions from the rivers. Particularly, the turbulence of

the river flow affected the variation of $CH_4$ the most among the three gases.



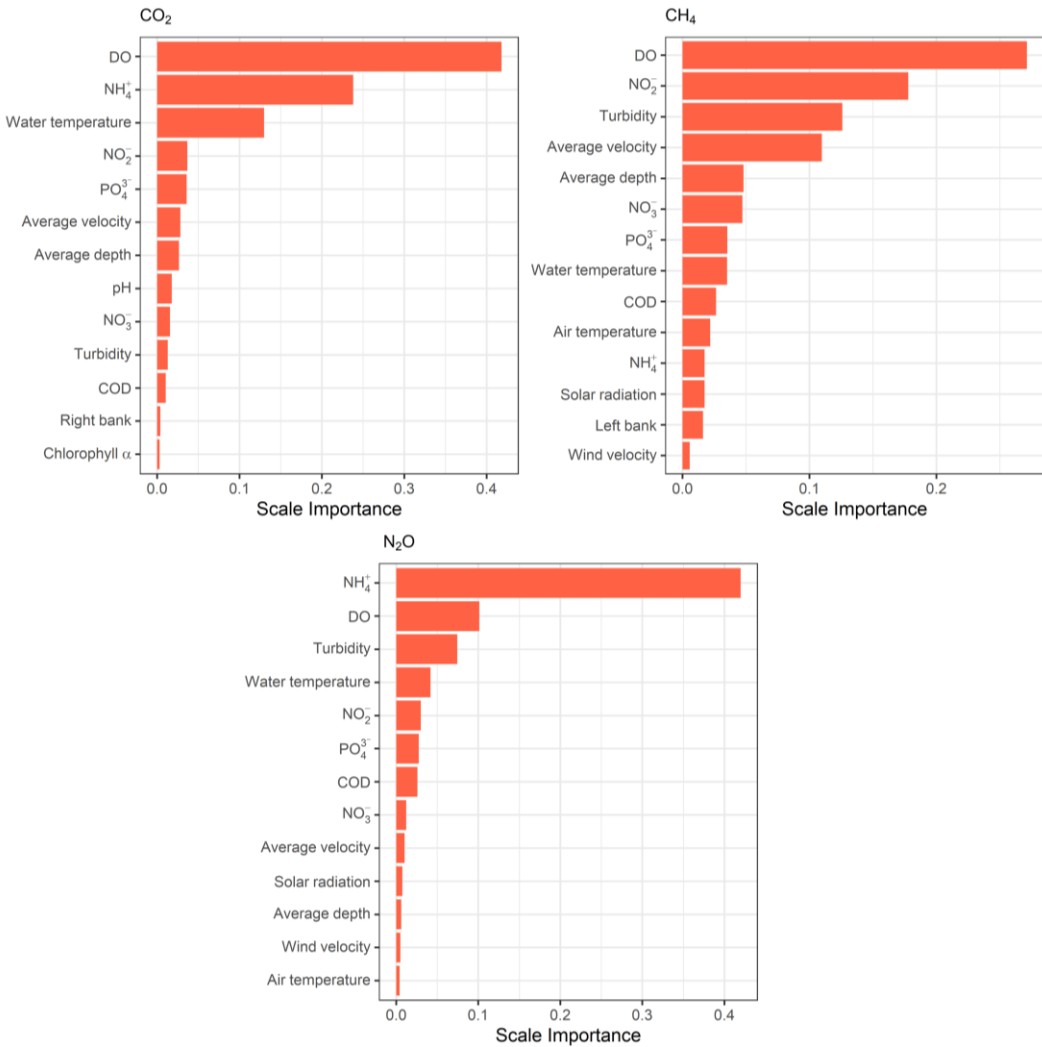

**Figure 5. Permutation accuracy importance of variables in the variation of GHG emissions from the Cuenca urban river system.**

**3.5 Sources of the GHG emissions**

Figure 6 shows the average of the estimated total GHG emissions per year from each tributary. Note that although the

standard error of the mean of the samples was used to indicate the uncertainty of the estimates, substantial variation of the

emissions can be induced by the fluctuation of water quality, and river and climatic conditions during a year. It appeared that

Tomebamba and Tarqui were the major contributors to GHG emissions in the Cuenca urban river system. Particularly,

Tomebamba tributary released the most with $186.8\pm51.8$ Gg $CO_2$ $yr^{-1}$, $1.5\pm0.6$ Gg $CH_4$ $yr^{-1}$, and $73.1\pm22.2$ Mg $N_2O$ $yr^{-1}$,

accounting for 57.0% of the total $CO_2$ emissions, 76.7% of the total $CH_4$ emissions, and 44.6% of the total $N_2O$ emissions

from the whole river basin. Details of the fraction of the total GHG emissions per year from different tributaries can be found





in Supplementary Material S7. High amount of GHG emissions from Tomebamba was caused by its high emission rate, i.e. $4527.9\pm1256.3$ mg-C m$^{-2}$ d$^{-1}$, $37.0\pm14.7$ mg-C m$^{-2}$ d$^{-1}$, and $1.78\pm0.53$ mg-N m$^{-2}$ d$^{-1}$ for $CO_2$, $CH_4$, and $N_2O$ emissions, respectively (Supplementary Material S7). These emission rates were more than double the mean GHG emission rates from the whole Cuenca urban river. The high emission rates were induced by the emissions of the sites 15, 16, and 35 in this

tributary. These contaminated sites, considered very heavily polluted by both indexes, had a high concentration of $NH_4^+$ of more than 15 mg L$^{-1}$ and a thick anaerobic sludge layer of 5-20 cm, causing low DO concentration of 2.7-3.4 mg L$^{-1}$. Note that due to the exclusion of ebullition in the flux calculation, the estimated $CH_4$ could be underestimated. High $CH_4$ emission rate of 3.04 mg-C m$^{-2}$ d$^{-1}$ also found in site 34 in Tomebamba tributary although we observed thin sludge layers and the highest wind velocity (5.8 m s$^{-1}$) in this site.

As the largest tributary, Tarqui ranked the second-highest contributor, releasing $110.4\pm48.9$ Gg $CO_2$ yr$^{-1}$, $0.5\pm0.2$ Gg $CH_4$ yr$^{-1}$, and $95.6\pm67.9$ Mg $N_2O$ yr$^{-1}$, accounting for 18.3%, 13.9%, and 31.5%, of the total emissions of $CO_2$, $CH_4$, and $N_2O$, respectively. The high value of the mean of the total $N_2O$ emissions from Tarqui was induced by an exceptionally high $N_2O$ emissions from site 02, i.e. 12.6 mg-N m$^{-2}$ d$^{-1}$. Note that this site was categorized in the worst water quality category by both indexes having the highest concentrations of TN, $NH_4$, turbidity, and TDS, together with low DO level of 5.54 mg L$^{-1}$. Site

22 was another polluted site in Tarqui tributary also characterized by very stagnant water with flow velocity of 0.01 m s$^{-1}$ and low DO level of 2.7 mg L$^{-1}$. This site was the second and the fifth largest contributor of $CO_2$ and $CH_4$, respectively. Conversely, being the least contaminated tributaries, Machangara and Yanuncay released less than 15 times the total GHG emissions from Tomebamba and Tarqui tributaries, in which each accounted for around 5% of the total $CO_2$ and $N_2O$ emissions, and 0.7% of total $CH_4$ emissions from the whole basin.

Note that despite being the smallest tributary, high GHG emissions were found in Cuenca tributary with $84.5\pm23.8$ Gg $CO_2$ yr$^{-1}$, $0.3\pm0.2$ Gg $CH_4$ yr$^{-1}$, and $37.2\pm9.4$ Mg $N_2O$ yr$^{-1}$, accounting for 13.1% of the total $CO_2$ emissions, 8.1% of the total $CH_4$ emissions, and 11.5% of the total $N_2O$ emissions from the whole river basin. The main reason for these high emissions can be because of the discharge from the Ucubamba WWTP to Cuenca tributary, raising the concentration of $NH_4^+$ and $PO_4^{3-}$ by five- and two-fold, respectively, i.e. above 5 mg $NH_4^+$-N L$^{-1}$ and 0.7 mg $PO_4^{3-}$-P L$^{-1}$. The high concentration of nutrients with

the overgrowth of algae were also found in the discharge of Ucubamba WWTP in the previous studies (Ho, 2018;Jerves-Cobo et al., 2020a). This result highlights the concern about the increasing GHG emissions from the WWTPs and the potential impact of its discharge on the GHG emissions from the receiving water bodies (Mannina et al., 2018).



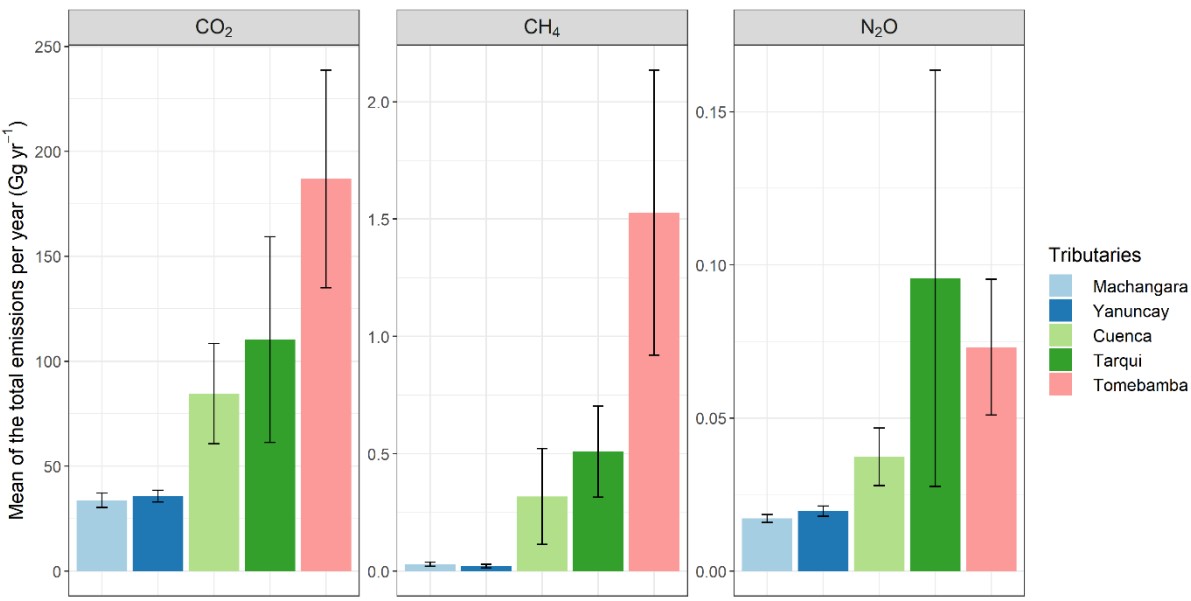

**Figure 6. Mean of the estimated total emissions per year from the five tributaries of the Cuenca urban river system. Error bars**
**represent the standard error of the mean of the sample.**

## 4. Conclusion

- Being the most polluted tributaries, Tomebamba and Tarqui, released 75% of the total emissions of $CO_2$ and $N_2O$, and 90% of the total $CH_4$ emissions, which was in contrast to the emissions from Machangara and Yanuncay, i.e. only 5% and 0.7%, respectively. High peaks of GHG emissions were found in the Cuenca tributary after the
discharge of the Ucubamba WWTP.

- By using Prati and Oregon Indexes, a clear pattern between water quality and GHG emissions was observed, in which the more polluted the sampling sites were, the higher were their GHG emissions. Specifically, when river water deteriorated from acceptable quality to very heavily polluted quality, their GWP increased by ten times. Compared to the estimated emissions from the global streams, rivers with polluted water can release almost double
the average estimated GWP while if their water quality worsened to very heavily polluted, the proportion was up to ten times. On the other hand, when the rivers had good water quality according to Prati Index, their GWP was only approximately half of the average estimated GWP while the GWP of acceptable-water-quality rivers was similar to this value. These results suggest that to estimate of the global emissions from inland waters, both their quantity and water quality should be considered for which Prati Index is recommended over the other.

- The study found that adjacent land-use types, i.e. urban, transportation systems, and agriculture, had significantly contributed to the increase in the GHG emissions from the rivers in Cuenca. Specifically, the GWP of the sites close



to urban areas was four time higher than the GWP of the sites close to natural areas. Similarly, the GWP of the sites close to roads or agricultural areas was triple and double the GWP of the natural sites. Note that the later was smaller than the average estimated GWP of global streams by 25%. These results highlight the indirect impacts of land-use change on increasing the GHG emissions from inland waters which are currently being omitted in land-use planning and resource management.

- By applying random forests, the main important factors on GHG emissions were identified. Dissolved $O_2$ appeared to be the most important factor for the variation of the $CO_2$ and $CH_4$ emissions and the second most important factor for the variation of the $N_2O$ emissions. Ammonium, together with variables indicating flow characteristics, such as turbidity, average velocity, average depth, and water temperature, also affected the variation of the GHG emissions. Conversely, a margin effect of organic matter concentration on the GHG emissions was found, which is in contrast to their strong correlation obtained from the previous studies. This result implies a higher role of (partial) nitrification compared to denitrification in producing $N_2O$ in these river systems.

**Acknowledgment**

This research was performed in the context of the VLIR Ecuador Biodiversity Network project. This project was funded by the Vlaamse Interuniversitaire Raad-Universitaire Ontwikkelingssamenwerking (VLIR-UOS), which supports partnerships between universities and university colleges in Flanders and the South. We thank Carlos Santiago Deluquez, Caio Neves, Paula Avila, Juan Enrique Orellana, and Kate Pesantez for their contributions during the sampling campaign. We are grateful to the Water and Soil Quality Analysis Laboratory of the University of Cuenca for their supports in our analyses.

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
