# Peer review of "Effects of land use and water quality on greenhouse gas emissions from an urban river system in Cuenca (Ecuador)"

_Biogeosciences, 2020_

## Referee Comment (RC1) · Anonymous Referee #1 · 16 Sep 2020

The manuscript entitled "Effects of land use and water quality on greenhouse gas emissions from an urban river system" provides data on GHG emissions from aquatic systems in a watershed located in Ecuador and investigates the link between water quality, adjacent land cover types and the magnitude of GHG emissions. The manuscript brings the importance of considering water quality on the estimates of the total GHG emissions from aquatic systems in addition to considering the total area only. This is a promising approach. However, there are many technical problems that need to be addressed. The estimation of gas transfer velocity (k) is largely discussed in the literature, and k estimates from empirical models should be used with caution. One major technical problem is that the gas fluxes were estimated using k600 parameterized as a function of wind speed, which is valid for open water systems, such as reservoirs, lakes

and oceans, but not recommended for rivers and streams. A flow-velocity- or water-depth-based model to estimate k600 should be considered as an alternative, and the associated uncertainties should be addressed. I suggest the authors to consider a recent paper published in the Biogeosciences by Li et al. 2019 and the cited papers to better estimate k. Additionally, the annual emissions were estimated using only data from 17/09/2018 to 21/09/2018 (5 days). This does not seem acceptable to me. There is no information about the number of samples per sampling site or any other information that justifies such extrapolation. In the manuscript, the authors mistakenly seem to use the Global Warming Potential (GWP) concept in the Results and Discussion section. Then, it is difficult to evaluate how authors estimated the emission of GHG in CO2-eq. In summary, the paper needs improvements on the method section, on the k estimates and the described data does not support most of the interpretations and conclusions. Personally, I support studies and papers that show results from tropical systems because of the relatively low available data and information. I encourage the authors to revise the paper with caution, but, unfortunately, in this form, the manuscript is not suitable for Biogeosciences.

Specific comments and suggestions are addressed below.

Ln 15 - Specify here that these indexes are water quality indexes.

Ln 23 - What authors mean with "nature sites"? Do you mean "sites close to forested areas"?

Ln 49-51 - The estimation of k600 is also a large challenge to estimate GHG emissions from aquatic systems. You should discuss k in the manuscript.

Ln 71 - The authors mentioned that the study area is 223 km2 but this value is less them the sum of the area of the studied sub-basins added in Ln 79. Please, verify, clarify or specify the study area.

Ln 81 - All data in the manuscript were collected in five days (from 17/09/2018 to

21/09/2018). Is it representative to discuss temporal variation? I agree that authors can evaluate temporal variation in a day scale approach. However, data from five consecutive days do not represent the annual variability and are not enough to estimate annual emissions of GHG.

Ln 83 - This sentence is not clear and there are many assumptions in the same sentence. Why do you assume that covering only daylight will ensure the investigation of temporal effects? Additionally, the connection between oxygen and GHG emission is not as simple as you stated in this sentence. Please, rephrase or remove this sentence.

Ln 90 - How many samples per sampling sites?? Were samples collected every day in each sampling site? Ln 95 - Please, change "Hack kit" to "Hach kit" in the Supplementary Material S1.

Ln 96 - Land use is one of the main subjects of the manuscript and it is also in the paper tittle. I suggest adding a subsection in the methods (as you did with the water quality) specifying what types of land use you considered, how they were determined and the characteristics of each land use (types of forests, agricultures, urban areas etc). Additionally, I suggest using the term "land use and land cover".

Ln 150 – Authors I suggest use a different symbol for partial pressure of the gas in the adjacent air.

Ln 157 - This sentence is not clear. The total watershed area should not be used for this calculation, but the water surface area should be used.

Ln 158-161 - There is a serious conceptual problem in this sentence. GWP is based on the capacity of a given gas to absorb heat compared to CO2. And here, the authors are assuming they are "determining" the GWP of the three gases. I assume that the authors are using GWP used on the Fifth Assessment Report to calculate emission in carbon dioxide equivalent (CO2-eq).

Ln 165-170 - The cited paper addressed lakes and pond, not streams and rivers. Additionally, this sentence is displaced in the text and should be removed from the method section. The following sentences showing some results should not be here in the method section. Please, remove.

Ln 174 - Why calculate two different indexes? I suggest using only one index. Why are you using Prati and Oregon indexes? The author stated that the Oregon index was developed to express ambient water quality for general recreational use. Are the aquatic systems in the watershed for recreation purposes? If yes, I suggest the authors to describe the multiuse purpose in the Study Area section.

Ln 223 - "total emissions of the three gases per year from the whole river basin". This statement is not supported by your data because you analyzed only five consecutive days of the year. Authors should consider use "during the sampling period" instead of "per year".

Ln 247-256 - This entire paragraph does not add any useful information to the manuscript. I suggest removing this paragraph. Additionally, I suggest using only one index and focus on the relationship between the index and GHG emissions.

Figure 3 – Please, add the number of samples (n) that compose each box. And, I cannot identify which class does not have any value in the Oregon Index graphs. I suggest insert the class names in the x axis of each panel.

Ln 261-266 - This is an important information and the authors do not need both indexes to have the same conclusion. As I suggested before, use only one index.

Ln 268 - GWP should not be used here in this sentence. Please see the comments in the methods section.

Ln 271-274 – Holgerson and Raymond (2016) do not address emission from streams, as the authors stated in this sentence. They estimated emissions from non-running inland water. Please, verify.

Table 2 - What is the difference between "Urban" and "Industry"? Are urban areas

residential areas? Please, specify each land use and land cover in the method section.

Figure 6 - This figure is in both the main text and Supplement Material. Please, remove from the SM.

[Figure]

---

## Referee Comment (RC2) · Anonymous Referee #2 · 4 Oct 2020

General comment: Ho and colleagues present a study on greenhouse gas emissions from a small urban river system in Ecuador. They evaluated the effects of water quality and land use types on the magnitude of GHG emission rates. Two WQIs were used to determine the water quality status and finally the random forest model was applied to identify the primary drivers for each of the three GHGs. Although land use and wastewater discharge have been widely recognized to impact GHG concentration and emissions worldwide, it has been seldom examined within a river system which encompass both human disturbances. Pollution of inland waters has become a global issue and how it has affected GHG emissions remains largely unknown. This study provides a timely investigation into this research question.

Major comments: 1) Sampling was conducted only once in late September, soon after

the start of the rainy season. I am not implying that this sampling strategy is wrong, but it is risky to use this 5-d sampling results to estimate the annual GHG emission flux. Considering the seasonality in hydrology, the calculated annual fluxes might have high uncertainties and don't really reflect the actual annual fluxes (and seasonal variations). Because of the focus of this study is to examine the impacts of land use types and water quality categories on GHG emissions, I would recommend the authors remove the calculation of the annual GHG emission fluxes and related discussion from the text. Please also refer to my other comments on sampling in the specific comments below.

2) Because estimation of the GHG flux involves dissolved GHG concentration and the gas transfer velocity across the water-air interface. For flowing rivers, the gas transfer velocity is more affected by flow velocity rather than by wind speed. The authors need to think about the suitability of using wind as a proxy for estimating the turbulence. Also, compared with emission flux, personally I think the dissolved GHG concentrations would be more appropriate as an indicator to examine the effects of land use and water quality.

Specific comments (with line number, L): L35: The rivers themselves are not the major sources of GHGs. Instead, a large proportion of the GHGs is derived from terrestrial ecosystems. L37: This flux has been updated. See Drake et al., 2018. Limnology and Oceanography Letter, 3, 132-142. L51/52: Given the spatial heterogeneity of hydrology, geomorphology, climate, etc, it is natural to see a strong spatial variation of GHG emissions. This has been widely observed worldwide. L62: There are a number of WQIs available for water quality assessment (see Zotou et al., 2019. Environ Monit Assess, 505), I am not clear how the authors chose these two indices for this comparison. It seems the Canadian Council of Ministers of Environment (CCME) index is more appropriate and common in describing water quality. L74/75: Over what period are these mean climate characteristics calculated? Please specify. L78: Sum of the fiver tributaries is not equal to the total drainage area of the basin (223 km2). L83: I don't know how the sampling during the daytime can ensure the investigation of temporal

effects. First, the sampling was only conducted during the daytime, thus the temporal effect of diel cycle cannot be detected. Second, the sampling was performed once only (soon after the start of the second rainy season of a year). Thus, the temporal effect of seasonal variation cannot be detected either. L101: how wide are the rivers? L120: Using the headspace equilibrium method to measure pCO2 with no consideration of alkalinity/DIC concentration is prone to errors and may lead to gross biases in the finally calculated pCO2 results. The authors can take a look on this recent open discussion (https://bg.copernicus.org/preprints/bg-2020-307/) and, if necessary, correct your calculations after headspace equilibration. L157: To calculate the total emissions of each river tributary, shouldn't it be based on the total stream surface area? I don't know how the total emission flux was calculated on the basis of total watershed area. If it is the total stream surface area used in the calculations, the authors need to elaborate on the details on the stream surface area estimation. L165: Holgerson and Raymond (2016) only looked at lakes (reservoirs), not at rivers and streams. Thus the results here are not directly comparable to Holgerson and Raymond (2016). L171: the authors need to justify why these two WQIs were used in this study. A brief justification will suffice. L222-225: how were these relative contributions calculated? Because the five tributary catchments have different catchment sizes, it is reasonable to observe a large contribution from the Tomebamba tributary because of its large catchment size. I don't think such comparison makes sense as it is not normalized by catchment area. In need, because almost all the sampling sites are located downstream near the catchment outlet, the calculated GHG emission fluxes are site-specific and don't really reflect the spatial variability of GHG emissions across the five tributaries or across the whole study area. L225-227: following my comment above, how was the spatial variation determined? By comparing the sites within each tributary catchment? This spatial variation is reasonably clear as the sites are located within different land use landscapes or with/without wastewater inputs. L230: The skewness was clearly caused the extremely high values. I am not sure if you should remove these outliers for this plotting. If the outliers are removed, the arithmetic means will be much lower. L263: this is a very good observation. Does the dissolved GHG concentration also show a similar variation by water quality level? Also, for the DO, does it also show a similar variation? Because GHG emission reflects the combined effect of dissolved GHG concentration and gas transfer velocity, it will be more meaningful to show the change of dissolved GHG centrations which are perhaps better indicative of the water quality status. L274: again, the cited values from Holgerson and Raymond, 2016 are for lakes only, not for rivers. L275: To estimate the global average GWP, perhaps you need to know the relative abundance of streams/rivers/lakes/reservoirs/wetlands. L291-303: Move these descriptions to the section 'study area'? L308: for the effect of land-use types on GHG emissions, I agree with the authors that land use will have diverse impacts on GHGH concentration and emissions. However, most of the sampling sites (Fig 1) are nested within the catchment. This suggests that the observed GHG emissions at these sites are not necessarily affected or controlled by only one single land use type. For example, the sites in the downstream may have been simultaneously affected by urban, agriculture and nature. If without an accurate quantification of their relative contributions, it is problematic to compare the impacts of different land use types on GHG emissions as shown in Fig 4. This is also related to the results in Table 2. Are these GWPs solely governed by one land use type? L368-370: how have thee factors affected the variation of the GHG emissions? Any evidence? L390: again, for these emission rates expressed as Gg $CO_2$ yr-1, I don't think it is reliable as expected. The sampling was not spatially and temporal resolved enough for an annual-scale estimation. L428: what does the 'later' refer to? I believe it is 'nature'. Fig 1. How did the authors differentiate rivers from streams? By Strahler order?

---

## Author Comment (AC1) · 5 Nov 2020

**Cover letter**

Manuscript ID: bg-2020-311

Effects of land use and water quality on greenhouse gas accumulation in an urban river system by Long Ho, Ruben Jerves-Cobo, Matti Barthel, Johan Six, Samuel Bode, Pascal Boeckx, and Peter Goethals.

Dear Editors and Reviewers,

We would like to thank the reviewers for their relevant and constructive remarks. We have revised our manuscript accordingly. We acknowledge that the revision, including the removal of flux calculation and the estimation of annual greenhouse gas (GHG) emissions from the rivers, abridged the manuscript and clarified the story of our research. Moreover, by removing these calculations, we also eliminated their uncertainty in our study while still reaching the same conclusions based on the dissolved GHG concentrations that were measured in the sampling campaigns. Other comments were also taken into account to further improve the manuscript.

We hope that the changes and explanations are acceptable and satisfactory with the expectation of the editors and reviewers. Below are their details. Note that the line numbers indicated below aim to clarify where the modifications can be found in the revised manuscript that can be provided upon the request of the editors and reviewers.

Thank you very much for your time and consideration!

Yours sincerely,

Long Ho

Department of Animal Sciences and Aquatic Ecology

Ghent University, Belgium

E-mail address: Long.TuanHo@Ugent.be.

**Referee #1**

The manuscript entitled "Effects of land use and water quality on greenhouse gas emissions from an urban river system" provides data on GHG emissions from aquatic systems in a watershed located in Ecuador and investigates the link between water quality, adjacent land cover types and the magnitude of GHG emissions. The manuscript brings the importance of considering water quality on the estimates of the total GHG emissions from aquatic systems in addition to considering the total area only. This is a promising approach. However, there are many technical problems that need to be addressed.

*Major comments*

**1. The estimation of gas transfer velocity (k)** is largely discussed in the literature, and k estimates from empirical models should be used with caution. One major technical problem is that the gas fluxes were estimated using k600 parameterized as a function of wind speed, which is valid for open water systems, such as reservoirs, lakes and oceans, but not recommended for rivers and streams. A flow-velocity- or water-depth-based model to estimate k600 should be considered as an alternative, and the associated uncertainties should be addressed. I suggest the authors to consider a recent paper published in the Biogeosciences by Li et al. 2019 and the cited papers to better estimate k.

**Authors' responses:**

We agree that the estimates of k600 values from empirical models should be carefully implemented. As indicated in Raymond et al. (2012), direct measurement is needed for an accurate estimate of gas exchange. Moreover, we recalculated the k600 values as a function of stream velocity, slope, and water depth via different fitted equations in Raymond et al. (2012); however, the obtained result of the different equations varied significantly. From that viewpoint, we removed the flux calculations and used the dissolved gas concentrations that we directly collected and measured in our sampling campaign. By doing so, we removed the uncertainty of the flux calculation as the measurement of the dissolved GHG required no additional assumptions. Despite this replacement, the conclusions of the study remained unchanged regarding the effects of land use and water quality on the accumulation of greenhouse gases in Cuenca urban river system. The modification of the abstract and conclusions in the revised manuscript can be found as follows.

**Abstract** (line 10-25)

- Rivers act as a natural source of greenhouse gases (GHGs). However, anthropogenic activities can largely alter the chemical composition and microbial communities of rivers, consequently affecting their GHG production. To investigate these impacts, we assessed the accumulation of $CO_2$, $CH_4$, and $N_2O$ in an urban river system (Cuenca, Ecuador). High variation of the dissolved GHG concentrations was found among river tributaries that mainly depended on water quality and neighboring landscapes. By using Prati and Oregon Water Quality Indexes, a clear pattern was observed between water quality and the GHG accumulation in which the more polluted the sites were, the higher were their dissolved GHG concentrations. When river water quality deteriorated from acceptable to very heavily polluted, the mean value of $pCO_2$ and dissolved $CH_4$ increased by up to ten times while this value of dissolved $N_2O$ was boosted by 15 times.

Furthermore, surrounding land-use types, i.e. urban, roads, and agriculture, significantly affected the GHG production in the rivers. Particularly, the average $pCO_2$ and dissolved $N_2O$ of the sites close to urban areas were almost four times higher than these values of the natural sites while this ratio was 25 times in case of dissolved $CH_4$. Lastly, by applying random forests, we identified dissolved oxygen, ammonium, and flow characteristics as the main important factors to the GHG productivity. Conversely, low impact of organic matter and nitrate concentration suggested a higher role of nitrification than denitrification in producing $N_2O$. These results highlighted the impacts of land-use types on the river emissions via water contamination by sewage discharges and surface runoff. Hence, to estimate the emissions from global streams, both their quantity and water quality should be included.

**Conclusion** (line 341-367)

- Being the most polluted tributary running through the city of Cuenca (Ecuador), Tomebamba contained four times higher amount of $pCO_2$ and dissolved $N_2O$ compared to the two purest tributaries, Machangara and Yanuncay, in the Cuenca urban river systems while this proportion was ten times higher in case of dissolved $CH_4$. Similarly, much higher dissolved GHG concentrations were also found in Tarqui and Cuenca tributaries which could be attributed to their high influx of nutrients and organic matter as a result of agricultural runoffs and WWTP discharges.

- A clear pattern between water quality and dissolved GHG concentrations was observed, in which the more polluted the sampling sites were, the higher were their dissolved GHG concentrations. Specifically, according to Prati Index, when river water quality worsened from acceptable to very heavily polluted, the mean concentration of $pCO_2$ and dissolved $CH_4$ rose by around 10 times while in case of dissolved $N_2O$, it was by 15 times. A similar, yet less obvious, trend was also found in case of Oregon Index. These results suggest that to estimate of the global emissions from inland waters, both their quantity and water quality should be considered for which Prati Index is recommended over the other.

- Adjacent land-use types, i.e. urban, transportation systems, and agriculture, had significantly contributed to the increase in the GHG accumulation in the rivers in Cuenca. Specifically, the mean value of $pCO_2$ and dissolved $N_2O$ increased fourfold from natural sites to urban sites while this ratio was 25 times in case of $CH_4$. Similarly, the average dissolved concentration of $CH_4$ increased by 10 and 20 times when the sites were surrounded by agricultural areas and roads, respectively, instead of natural forests. These results highlighted the indirect impacts of land-use and land cover change on increasing GHG production from inland waters which are currently being omitted in land-use planning and resource management.

- The main important factors on dissolved GHG concentrations were identified by the application of random forests. Dissolved $O_2$ appeared to be the most important factor for the variation of the $pCO_2$ and dissolved $CH_4$ and the second most important factor for the variation of the dissolved $N_2O$. Ammonium, together with variables indicating flow characteristics, such as turbidity, average velocity, average depth, and water temperature, also affected the variation of the dissolved GHG concentrations. Conversely, a margin effect of organic matter concentration

was found, which is in contrast to their strong correlation obtained from the previous studies. This result implies a higher role of (partial) nitrification compared to denitrification in producing $N_2O$ in these river systems.

**2. The annual emissions were estimated using only data from 17/09/2018 to 21/09/2018 (5 days).** This does not seem acceptable to me. There is no information about the number of samples per sampling site or any other information that justifies such extrapolation

**Authors' responses:**

We agree that the data of our sampling campaign is insufficient for estimation of the annual emissions given the temporal effect of seasonal variation and whole diurnal cycle. Besides, as we removed the flux estimations as indicated in our response to the first major comment, we also removed the estimation of the annual emissions in the revised manuscript. This removal abridged the manuscript and condensed the research findings to the effects of land-use changes and water quality on the GHG accumulation in an urban river system.

**3. In the manuscript, the authors mistakenly seem to use the Global Warming Potential (GWP)** concept in the Results and Discussion section. Then, it is difficult to evaluate how authors estimated the emission of GHG in CO2-eq.

**Authors' responses:**

We had used the GWP to convert the $CH_4$ and $N_2O$ emissions to mg-$CO_2$ equivalent $m^{-2}$ $d^{-1}$. The values had been extracted from the Fifth Assessment Report by the Intergovernmental Panel on Climate Change (IPCC) 2015. This is the most recent report of IPCC. However, we realized that this conversion did not add to the story, we excluded this conversion in the revised manuscript to make the story of the research more concise and clear. Therefore, we used only the non-converted direct concentration data.

**4. In summary, the paper needs improvements on the method section**, on the k estimates and the described data does not support most of the interpretations and conclusions.

**Authors' responses:**

As mentioned in our response to the first major comment, the k600 estimate was removed in the revised manuscript. As such, no description of this estimation is needed. On the other hand, we agree on the addition of a description of land use and land cover in the Materials and Methods section as indicated in our response to the second and eighth specific comments.

*Specific comments*

**1. Ln 15** - Specify here that these indexes are water quality indexes.

Added.

**2. Ln 23** - What authors mean with "nature sites"? Do you mean "sites close to forested areas"?

The explanation of nature sites was included in the new section **2.5 Land use and land cover** in the Materials and Methods section as follows.

- To evaluate the effects of land use on the dissolved GHG concentrations of the tributaries, we considered five different types of land use, i.e. nature (close to the forests), industry (close to factories, and mining areas), agriculture (close to arable land, orchard, and farms), roads, and urban areas (close to residential and urban areas). (line 153-156)

**3. Ln 49-51** - The estimation of k600 is also a large challenge to estimate GHG emissions from aquatic systems. You should discuss k in the manuscript.

The flux calculation was removed; hence the discussion of $k_{600}$ values is not necessary. This helped to keep the story of the research focused and concise.

**4. Ln 71** - The authors mentioned that the study area is 223 km2 but this value is less them the sum of the area of the studied sub-basins added in Ln 79. Please, verify, clarify or specify the study area.

This info had been incorrect in the original manuscript and was adjusted in the revised manuscript as follows.

- The study area is 572.92 $km^2$, representing 25% of the Cuenca River basin (Figure S2.1). (line 70)

Moreover, a figure was added in the revised supplementary material to indicate the study area under the scope of the whole Cuenca river basin.

[Figure]

Figure S2.1. Study area in the scope of the whole Cuenca river basin.

**5. Ln 81** - All data in the manuscript were collected in five days (from 17/09/2018 to 21/09/2018). Is it representative to discuss temporal variation? I agree that authors can evaluate temporal variation in a day scale approach. However, data from five consecutive days do not represent the annual variability and are not enough to estimate annual emissions of GHG.

As mentioned in our response to the second major comment, we agree that our data were insufficient to calculate the annual variability of the GHG concentrations in the Cuenca river basin. Hence, we removed this sentence in the revised manuscript.

**6. Ln 83** - This sentence is not clear and there are many assumptions in the same sentence. Why do you assume that covering only daylight will ensure the investigation of temporal effects? Additionally, the connection between oxygen and GHG emission is not as simple as you stated in this sentence. Please, rephrase or remove this sentence.

We removed this sentence in the revised manuscript as mentioned in the previous comment.

**7. Ln 90** - How many samples per sampling sites?? Were samples collected every day in each sampling site? Ln 95 - Please, change "Hack kit" to "Hach kit" in the Supplementary Material S1.

Changed.

**8. Ln 96** - Land use is one of the main subjects of the manuscript and it is also in the paper tittle. I suggest adding a subsection in the methods (as you did with the water quality) specifying what types of land use you considered, how they were determined and the characteristics of each land use (types of forests, agricultures, urban areas etc). Additionally, I suggest using the term "land use and land cover".

We agree that details of the land use and land cover should be introduced, hence we added a new subsection 2.5 Land use and land cover in the Materials and Methods as follows.

- Due to the large sampling area, land-use types widely varied while other hydro-morphological variables remained relatively stable across the five tributaries. Particularly, urban and resident areas were dominant with around 55% of the total sampling areas, while forest and agriculture occupied 8-11% and 14-20%, respectively. Minor sampling area was surrounded by industrial factories and construction sites, with less than 5% each. Several riversides were next to the road, occupying 11-19% of the total sampling area. The distribution of the land-use types was not evenly among the rivers. Intensive urban activities can be found near the Cuenca and from the middle to the end of Tomebamba rivers. Conversely, Yanuncay and Machangara cross two natural reserves, i.e. Cajas National Park and the Machangara-Tomebamba protected forest, leading to their pristine water quality conditions. In addition, these two aforementioned rivers cross the city of Cuenca in the latest part of their path before the confluence with the Tomebamba River (Jerves-Cobo et al., 2018). In addition, these two aforementioned rivers cross the city of Cuenca in the latest part of their path before the confluence with the Tomebamba River. Tarqui river locates near agricultural irrigation and livestock production areas, causing their high nutrient and organic inputs (Jerves-Cobo et al., 2018). To evaluate the effects of land use on the dissolved GHG concentrations of the tributaries, we considered five different types of land use, i.e. nature (close to the forests), industry (close to factories, and mining areas),

agriculture (close to arable land, orchard, and farms), roads, and urban areas (close to residential and urban areas). (line 145-156)

**9. Ln 150** – Authors I suggest use a different symbol for partial pressure of the gas in the adjacent air.

As mentioned in the previous comments, the flux calculation including this part was removed.

**10. Ln 157** - This sentence is not clear. The total watershed area should not be used for this calculation, but the water surface area should be used.

As the calculation of the fluxes was removed, the calculation of the total annual emissions per tributary was also removed.

**11. Ln 158-161** - There is a serious conceptual problem in this sentence. GWP is based on the capacity of a given gas to absorb heat compared to CO2. And here, the authors are assuming they are "determining" the GWP of the three gases. I assume that the authors are using GWP used on the Fifth Assessment Report to calculate emission in carbon dioxide equivalent (CO2-eq)

As mentioned in our response to the third major comment, we realized that this conversion did not add to the story, thus we excluded this conversion in the revised manuscript to make the story of the research more focused and concise. We used only the results obtained from our sampling campaign instead to avoid the confusion of the readers.

**12. Ln 165-170** - The cited paper addressed lakes and pond, not streams and rivers. Additionally, this sentence is displaced in the text and should be removed from the method section. The following sentences showing some results should not be here in the method section. Please, remove.

Removed.

**13. Ln 174** - Why calculate two different indexes? I suggest using only one index. Why are you using Prati and Oregon indexes? The author stated that the Oregon index was developed to express ambient water quality for general recreational use. Are the aquatic systems in the watershed for recreation purposes? If yes, I suggest the authors to describe the multiuse purpose in the Study Area section.

The justification of the two indexes was added in the revised manuscript as follows.

- By aggregating the measurements of multiple water quality parameters, WQI as a single number can be used to assess the quality of a water resource for serving different purposes (Lumb et al., 2011). However, as each WQI has its own demerits, no WQIs can be universally applicable (Tyagi et al., 2013). Among 30 WQIs listed in Sutadian et al. (2016), Prati and Oregon indexes were chosen in this study because of their successful establishment for river water quality assessment (Zotou et al., 2019) and their required parameters were measured during the sampling campaigns. Other WQIs, such as National Sanitation Foundation (NSF) WQI or Stoner's index, need parameters that were not measured in this study. Additionally, despite their suitable required parameters, Canadian Council of Ministers of Environment (CCME) index and Weighted Arithmetic WQI were not applied due to their common application in drinking water use (Sutadian et al., 2016) and their limitations listed in Tyagi et al. (2013). (line 126-134)

Similar to Prati Index, Oregon Index has been applied not only in Oregon but also in many places in the world. In fact, it is one of the most important water quality indices according to several review articles on water quality indexes, such as Lumb et al. (2011), Sutadian et al. (2016), and Zotou et al. (2019). We added the review article to illustrate the justification of the two indexes as follows.

- Prati index, developed by Prati et al. (1971), is often used to evaluate surface water quality with a consideration of numerous pollutants. Oregon Index was developed by Dunnette (1979) and then modified by Cude (2001) to initially express ambient water quality for general recreational use. After that, both indexes have widely been applied to assess river water quality for different purposes (Lumb et al., 2011;Sutadian et al., 2016;Zotou et al., 2019). (line 135-138)

**14. Ln 223** - "total emissions of the three gases per year from the whole river basin". This statement is not supported by your data because you analyzed only five consecutive days of the year. Authors should consider use "during the sampling period" instead of "per year".

As mentioned in our response to the third major comment, we realized that this conversion did not add to the story, thus the estimation of total annual emission from the tributaries was removed.

**15. Ln 247-256** - This entire paragraph does not add any useful information to the manuscript. I suggest removing this paragraph. Additionally, I suggest using only one index and focus on the relationship between the index and GHG emissions

This paragraph aimed to analyze and compare the obtained results of the two indexes. Particularly, the river water quality was categorized more consistently in case of Prati Index compared to Oregon Index which can be attributed to the heavy penalty for high concentrations of organic matter and nutrients in case of the latter. In fact, this led to only two sites of the Cuenca river basin were considered either good or fair water quality. The categorization based on Prati Index was more in line with the results of the previous sampling campaigns of Jerves-Cobo et al. (2018) and Jerves-Cobo et al. (2020). Because of these reasons, together with its simple calculation, Prati Index was recommended over Oregon Index in our case study. This was illustrated in the revised manuscript as follows.

- Prati and Oregon Indexes were applied to assess the effects of water quality on the dissolved GHG concentrations in the Cuenca river basin. According to the Prati Index, the rivers had higher water quality than the results obtained from the Oregon Index. Particularly, 18 sampling sites were categorized in either good quality or acceptable quality following the Prati Index while only two sites were considered either good or fair water quality according to the Oregon Index. The results obtained by using Prati Index appeared to be more in line with the results obtained from the previous sampling campaigns of Jerves-Cobo et al. (2018) and Jerves-Cobo et al. (2020). (line 233-238)

Regarding the justification of the two WQIs, since each WQI has its own demerits, no WQIs can be universally applicable (Tyagi et al., 2013). The two indexes were chosen among 30 WQIs listed in Sutadian et al. (2016) because of their successful establishment for river water quality assessment (Zotou et al., 2019) and their required parameters were measured during the sampling campaigns. The justification of the two WQIs was added in the revised manuscript as follows.

- By aggregating the measurements of multiple water quality parameters, WQI as a single number can be used to assess the quality of a water resource for serving different purposes (Lumb et al., 2011). However, as each WQI has its own demerits, no WQIs can be universally applicable (Tyagi et al., 2013). Among 30 WQIs listed in Sutadian et al. (2016), Prati and Oregon indexes were chosen in this study because of their successful establishment for river water quality assessment (Zotou et al., 2019) and their required parameters were measured during the sampling campaigns. Other WQIs, such as National Sanitation Foundation (NSF) WQI or Stoner's index, need parameters that were not measured in this study. Additionally, despite their suitable required parameters, Canadian Council of Ministers of Environment (CCME) index and Weighted Arithmetic WQI were not applied due to their common application in drinking water use (Sutadian et al., 2016) and their limitations listed in Tyagi et al. (2013). (line 126-134)

**16. Figure 3** – Please, add the number of samples (n) that compose each box. And, I cannot identify which class does not have any value in the Oregon Index graphs. I suggest insert the class names in the x axis of each panel.

The name of the WQ categories was added to the x-axis while the number of samples was put in Table S3.1 to avoid making this figure complicated.

[Figure]

Figure 3. Dissolved concentrations of the three greenhouse gases from the Cuenca urban river system in different water quality categories using Oregon and Prati Indexes. Box plots display 10th, 25th, 50th, 75th and 90th percentiles, and individual data points outside the 10th and 90th percentiles. Blue dots represent the mean of the dissolved concentrations in the water quality categories.

Table S3.1 Number of sites in each water quality category in both indexes

| Water Quality Category (Prati/Oregon Index) | Number of sites Prati Index | Number of sites Oregon Index |
|---|---|---|
| Good/ Excellent | 6 | 0 |
| Acceptable/ Good | 12 | 1 |
| Polluted/ Fair | 9 | 1 |
| Heavily Polluted/ Poor | 6 | 10 |
| Very Heavily Polluted/ Very Poor | 3 | 24 |

**17. Ln 261-266** - This is an important information and the authors do not need both indexes to have the same conclusion. As I suggested before, use only one index.

As mentioned in the previous responses, the justification of the two indexes was described in the revised manuscript.

**18. Ln 268** - GWP should not be used here in this sentence. Please see the comments in the methods section.

As mentioned in the previous responses, GWP part was removed in the revised manuscript.

**19. Ln 271-274** – Holgerson and Raymond (2016) do not address emission from streams, as the authors stated in this sentence. They estimated emissions from non-running inland water. Please, verify

We removed this reference.

**20. Table 2** - What is the difference between "Urban" and "Industry"? Are urban areas residential areas? Please, specify each land use and land cover in the method section.

We clarify this aspect in the new section *2.5 Land use and land cover* in the Materials and Methods as follows.

- To evaluate the effects of land use on the dissolved GHG concentrations of the tributaries, we considered five different types of land use, i.e. nature (close to the forests), industry (close to factories, and mining areas), agriculture (close to arable land, orchard, and farms), roads, and urban areas (close to residential and urban areas). (line 153-156)

**21. Figure 6** - This figure is in both the main text and Supplement Material. Please, remove from the SM.

We removed this figure since the estimation of the total annual emissions from the tributaries was removed.

**References**

Cude, C. G.: Oregon Water Quality Index: A tool for evaluating water quality management effectiveness, J Am Water Resour As, 37, 125-137, DOI 10.1111/j.1752-1688.2001.tb05480.x, 2001.

Dunnette, D.: A geographically variable water quality index used in Oregon, Journal (Water Pollution Control Federation), 53-61, 1979.

Jerves-Cobo, R., Lock, K., Van Butsel, J., Pauta, G., Cisneros, F., Nopens, I., and Goethals, P. L. M.: Biological impact assessment of sewage outfalls in the urbanized area of the Cuenca River basin (Ecuador) in two different seasons, Limnologica, 71, 8-28, 10.1016/j.limno.2018.05.003, 2018.

Jerves-Cobo, R., Forio, M. A. E., Lock, K., Van Butsel, J., Pauta, G., Cisneros, F., Nopens, I., and Goethals, P. L. M.: Biological water quality in tropical rivers during dry and rainy seasons: A model-based analysis, Ecological Indicators, 108, UNSP 105769

10.1016/j.ecolind.2019.105769, 2020.

Lumb, A., Sharma, T. C., and Bibeault, J. F.: A Review of Genesis and Evolution of Water Quality Index (WQI) and Some Future Directions, Water Qual Expos Hea, 3, 11-24, 10.1007/s12403-011-0040-0, 2011.

Prati, L., Pavanello, R., and Pesarin, F.: Assessment of Surface Water Quality by a Single Index of Pollution, Water Res, 5, 741-+, Doi 10.1016/0043-1354(71)90097-2, 1971.

Raymond, P. A., Zappa, C. J., Butman, D., Bott, T. L., Potter, J., Mulholland, P., Laursen, A. E., McDowell, W. H., and Newbold, D.: Scaling the gas transfer velocity and hydraulic geometry in streams and small rivers, Limnology and Oceanography: Fluids and Environments, 2, 41-53, 10.1215/21573689-1597669, 2012.

Sutadian, A. D., Muttil, N., Yilmaz, A. G., and Perera, B. J. C.: Development of river water quality indices-a review, Environ Monit Assess, 188, 10.1007/s10661-015-5050-0, 2016.

Tyagi, S., Sharma, B., Singh, P., and Dobhal, R.: Water Quality Assessment in Terms of Water Quality Index, American Journal of Water Resources, 1, 34-38, 2013.

Zotou, I., Tsihrintzis, V. A., and Gikas, G. D.: Performance of Seven Water Quality Indices (WQIs) in a Mediterranean River, Environ Monit Assess, 191, 10.1007/s10661-019-7652-4, 2019.

---

## Author Comment (AC2) · 5 Nov 2020

**Cover letter**

Manuscript ID: bg-2020-311

Effects of land use and water quality on greenhouse gas accumulation in an urban river system by Long Ho, Ruben Jerves-Cobo, Matti Barthel, Johan Six, Samuel Bode, Pascal Boeckx, and Peter Goethals.

Dear Editors and Reviewers,

We would like to thank the reviewers for their relevant and constructive remarks. We have revised our manuscript accordingly. We acknowledge that the revision, including the removal of flux calculation and the estimation of annual greenhouse gas (GHG) emissions from the rivers, abridged the manuscript and clarified the story of our research. Moreover, by removing these calculations, we also eliminated their uncertainty in our study while still reaching the same conclusions based on the dissolved GHG concentrations that were measured in the sampling campaigns. Other comments were also taken into account to further improve the manuscript.

We hope that the changes and explanations are acceptable and satisfactory with the expectation of the editors and reviewers. Below are their details. Note that the line numbers indicated below aim to clarify where the modifications can be found in the revised manuscript that can be provided upon the request of the editors and reviewers.

Thank you very much for your time and consideration!

Yours sincerely,

Long Ho

Department of Animal Sciences and Aquatic Ecology

Ghent University, Belgium

E-mail address: Long.TuanHo@Ugent.be.

**Referee #2**

Ho and colleagues present a study on greenhouse gas emissions from a small urban river system in Ecuador. They evaluated the effects of water quality and land use types on the magnitude of GHG emission rates. Two WQIs were used to determine the water quality status and finally the random forest model was applied to identify the primary drivers for each of the three GHGs. Although land use and wastewater discharge have been widely recognized to impact GHG concentration and emissions worldwide, it has been seldom examined within a river system which encompass both human disturbances. Pollution of inland waters has become a global issue and how it has affected GHG emissions remains largely unknown. This study provides a timely investigation into this research question.

*Major comments*

**1.** Sampling was conducted only once in late September, soon after the start of the rainy season. I am not implying that this sampling strategy is wrong, but it is risky to use this 5-d sampling results to estimate the annual GHG emission flux. Considering the seasonality in hydrology, the calculated annual fluxes might have high uncertainties and don't really reflect the actual annual fluxes (and seasonal variations). Because of the focus of this study is to examine the impacts of land use types and water quality categories on GHG emissions, I would recommend the authors remove the calculation of the annual GHG emission fluxes and related discussion from the text.

We agree that the data of our sampling campaign is insufficient for estimation of the annual emissions given the temporal effect of seasonal variation and whole diurnal cycle. Besides, as we removed the flux estimations, we also removed the estimation of the annual emissions in the revised manuscript. This removal abridged the manuscript and condensed the research findings to the effects of land-use changes and water quality on the GHG accumulation in an urban river system.

**2.** Because estimation of the GHG flux involves dissolved GHG concentration and the gas transfer velocity across the water-air interface. For flowing rivers, the gas transfer velocity is more affected by flow velocity rather than by wind speed. The authors need to think about the suitability of using wind as a proxy for estimating the turbulence. Also, compared with emission flux, personally I think the dissolved GHG concentrations would be more appropriate as an indicator to examine the effects of land use and water quality.

We agree that the estimates of k600 values from empirical models should be carefully implemented. As indicated in Raymond et al. (2012), direct measurement is needed for an accurate estimate of gas exchange. Moreover, we recalculated the k600 values as a function of stream velocity, slope, and water depth via different fitted equations in Raymond et al. (2012); however, the obtained result of the different equations varied significantly. From that viewpoint, we removed the flux calculations and used the dissolved gas concentrations that we directly collected and measured in our sampling campaign. By doing so, we removed the uncertainty of the flux calculation as the measurement of the dissolved GHG required no additional assumptions. Despite this replacement, the conclusions of the study remained unchanged regarding the effects of land use and water quality on the accumulation of greenhouse gases

in Cuenca urban river system. The modification of the abstract and conclusions in the revised manuscript can be found as follows.

**Abstract** (line 10-25)

- Rivers act as a natural source of greenhouse gases (GHGs). However, anthropogenic activities can largely alter the chemical composition and microbial communities of rivers, consequently affecting their GHG production. To investigate these impacts, we assessed the accumulation of $CO_2$, $CH_4$, and $N_2O$ in an urban river system (Cuenca, Ecuador). High variation of the dissolved GHG concentrations was found among river tributaries that mainly depended on water quality and neighboring landscapes. By using Prati and Oregon Water Quality Indexes, a clear pattern was observed between water quality and the GHG accumulation in which the more polluted the sites were, the higher were their dissolved GHG concentrations. When river water quality deteriorated from acceptable to very heavily polluted, the mean value of $pCO_2$ and dissolved $CH_4$ increased by up to ten times while this value of dissolved $N_2O$ was boosted by 15 times. Furthermore, surrounding land-use types, i.e. urban, roads, and agriculture, significantly affected the GHG production in the rivers. Particularly, the average $pCO_2$ and dissolved $N_2O$ of the sites close to urban areas were almost four times higher than these values of the natural sites while this ratio was 25 times in case of dissolved $CH_4$. Lastly, by applying random forests, we identified dissolved oxygen, ammonium, and flow characteristics as the main important factors to the GHG productivity. Conversely, low impact of organic matter and nitrate concentration suggested a higher role of nitrification than denitrification in producing $N_2O$. These results highlighted the impacts of land-use types on the river emissions via water contamination by sewage discharges and surface runoff. Hence, to estimate the emissions from global streams, both their quantity and water quality should be included.

**Conclusion** (line 341-367)

- Being the most polluted tributary running through the city of Cuenca (Ecuador), Tomebamba contained four times higher amount of $pCO_2$ and dissolved $N_2O$ compared to the two purest tributaries, Machangara and Yanuncay, in the Cuenca urban river systems while this proportion was ten times higher in case of dissolved $CH_4$. Similarly, much higher dissolved GHG concentrations were also found in Tarqui and Cuenca tributaries which could be attributed to their high influx of nutrients and organic matter as a result of agricultural runoffs and WWTP discharges.

- A clear pattern between water quality and dissolved GHG concentrations was observed, in which the more polluted the sampling sites were, the higher were their dissolved GHG concentrations. Specifically, according to Prati Index, when river water quality worsened from acceptable to very heavily polluted, the mean concentration of $pCO_2$ and dissolved $CH_4$ rose by around 10 times while in case of dissolved $N_2O$, it was by 15 times. A similar, yet less obvious, trend was also found in case of Oregon Index. These results suggest that to estimate of the global emissions from inland waters, both their quantity and water quality should be considered for which Prati Index is recommended over the other.

- Adjacent land-use types, i.e. urban, transportation systems, and agriculture, had significantly contributed to the increase in the GHG accumulation in the rivers in Cuenca. Specifically, the mean value of $pCO_2$ and dissolved $N_2O$ increased fourfold from natural sites to urban sites while this ratio was 25 times in case of $CH_4$. Similarly, the average dissolved concentration of $CH_4$ increased by 10 and 20 times when the sites were surrounded by agricultural areas and roads, respectively, instead of natural forests. These results highlighted the indirect impacts of land-use and land cover change on increasing GHG production from inland waters which are currently being omitted in land-use planning and resource management.
- The main important factors on dissolved GHG concentrations were identified by the application of random forests. Dissolved $O_2$ appeared to be the most important factor for the variation of the $pCO_2$ and dissolved $CH_4$ and the second most important factor for the variation of the dissolved $N_2O$. Ammonium, together with variables indicating flow characteristics, such as turbidity, average velocity, average depth, and water temperature, also affected the variation of the dissolved GHG concentrations. Conversely, a margin effect of organic matter concentration was found, which is in contrast to their strong correlation obtained from the previous studies. This result implies a higher role of (partial) nitrification compared to denitrification in producing $N_2O$ in these river systems.

**Specific comments**

**1. L35**: The rivers themselves are not the major sources of GHGs. Instead, a large proportion of the GHGs is derived from terrestrial ecosystems

We rephrased the sentence accordingly as follows

- Particularly, $CO_2$ and $CH_4$ are released via the decay of organic matter during bacterial decomposition processes while nitrifying and denitrifying microorganisms are considered generators of $N_2O$ in inland water bodies (Daelman et al., 2013). Besides acting as a natural source of GHGs, rivers also serve as conduits for the GHGs released from soil pore water, groundwater and sediments to the atmosphere as a result of substantial terrestrial-to-aquatic C flux of 5.1 Pg C $yr^{-1}$ (Hotchkiss et al., 2015;Drake et al., 2018). (line 30-34)

**2. L37**: This flux has been updated. See Drake et al., 2018. Limnology and Oceanography Letter, 3, 132-142.

We updated the new estimation from Drake et al., 2018 as follows

- In total, it was estimated from global streams and rivers that their $CO_2$ emissions were 3.9 Pg C $yr^{-1}$ (Drake et al., 2018) (line 34-35)

**3. L51/52**: Given the spatial heterogeneity of hydrology, geomorphology, climate, etc, it is natural to see a strong spatial variation of GHG emissions. This has been widely observed worldwide.

We rephrased this sentence following your comments as follows

- The challenges derive from the complex biological processes in the water column of rivers, their intricate interactions with terrestrial ecosystems and various human activities along the rivers.

Together with the biochemical complexity, the heterogeneity of river hydrology, geomorphology, climate and status induces substantial spatial variation of GHG emissions (Musenze et al., 2014;Borges et al., 2015;Qu et al., 2017). (line 48-51)

**4. L62**: There are a number of WQIs available for water quality assessment (see Zotou et al., 2019. Environ Monit Assess, 505), I am not clear how the authors chose these two indices for this comparison. It seems the Canadian Council of Ministers of Environment (CCME) index is more appropriate and common in describing water quality.

Indeed, there are many WQIs available for different purposes. Among 30 WQIs listed in Sutadian et al. (2016), we chose the two indexes because of their well establishment for river water quality assessment (Zotou et al., 2019) and their required parameters were measured during the sampling campaigns. Other WQIs, such as National Sanitation Foundation (NSF) WQI or Stoner's index, require parameters that were not measured in this study, while despite their suitable required parameters, Canadian Council of Ministers of Environment (CCME) index and Weighted Arithmetic WQI were not applied due to their common application in drinking water use (Sutadian et al., 2016) and their limitations listed in Tyagi et al. (2013). Particularly, Tyagi et al. (2013) listed 10 limitations of CCME as follows:

1. Loss of information on single variables.

2. Loss of information about the objectives specific to each location and particular water use.

3. Sensitivity of the results to the formulation of the index.

4. Loss of information on interactions between variables.

5. Lack of portability of the index to different ecosystem types.

6. Easy to manipulate (biased).

7. The same importance is given to all variables.

8. No combination with other indicators or biological data

9. Only partial diagnostic of the water quality.

10. F1not (one of the components in the CCME calculation) working appropriately when too few variables are considered or when too much covariance exists among them

Considering these viewpoints, we excluded CCME index in our study. The justification of the two indexes were added in the revised manuscript as follows.

- By aggregating the measurements of multiple water quality parameters, WQI as a single number can be used to assess the quality of a water resource for serving different purposes (Lumb et al., 2011). However, as each WQI has its own demerits, no WQIs can be universally applicable (Tyagi et al., 2013). Among 30 WQIs listed in Sutadian et al. (2016), Prati and Oregon indexes were chosen in this study because of their successful establishment for river water quality assessment (Zotou et al., 2019) and their required parameters were measured during the sampling campaigns. Other WQIs, such as National Sanitation Foundation (NSF) WQI or Stoner's index, need parameters that were not measured in this study. Additionally, despite their

suitable required parameters, Canadian Council of Ministers of Environment (CCME) index and Weighted Arithmetic WQI were not applied due to their common application in drinking water use (Sutadian et al., 2016) and their limitations listed in Tyagi et al. (2013). (line 126-134)

**5. L74/75**: Over what period are these mean climate characteristics calculated? Please specify.

The values were estimated based on the records from 1977 to 2011 from two meteorological stations located in the Mariscal Lamar Airport of Cuenca and the University of Cuenca, which is near to the Tomebamba river. This info was added to the revised manuscript as follows.

- The annual average air temperature is 16.3 °C and the average rainfall is about 879 mm per year. The values were estimated based on the records from 1977 and 2011 from the meteorological station located in the Mariscal Lamar Airport, which is in the urban area of Cuenca, as well as with information from 2001 to 2009 from the metrological station of the University of Cuenca, which is near sites 15 and 16 in Figure 1 (Jerves-Cobo et al., 2018a) (line 74-75)

**6. L78**: Sum of the river tributaries is not equal to the total drainage area of the basin (223 km2).

This info had been incorrect in the original manuscript and was adjusted in the revised manuscript as follows.

- The study area is 572.92 km$^2$, representing 25% of the Cuenca River basin (Figure S2.1). (line 70)

Moreover, a figure was added in the revised supplementary material to indicate the study area under the scope of the whole Cuenca river basin

[Figure]

Figure S2.1. Study area in the scope of the whole Cuenca river basin.

**7. L83**: I don't know how the sampling during the daytime can ensure the investigation of temporal effects. First, the sampling was only conducted during the daytime, thus the temporal effect of diel cycle cannot be detected. Second, the sampling was performed once only (soon after the start of the second rainy season of a year). Thus, the temporal effect of seasonal variation cannot be detected either.

We agree that the data of our sampling campaign is insufficient for estimation of the annual emissions given the temporal effect of seasonal variation and whole diurnal cycle. As such, we removed this sentence in our revised manuscript.

**8. L101**: how wide are the rivers?

We did not measure the width of the rivers. However, from the previous sampling campaigns of Jerves-Cobo et al. (2018b) and Jerves-Cobo et al. (2020), the width of the rivers varies significantly. Details of the width range of the rivers were added as follows.

- The area of Cuenca, Machangara, Tarqui, Tomebamba, and Yanuncay is 95.92, 111.19, 138.98, 113.03, 113.81 km$^2$, respectively and their width range is 2.8-29.7, 3.5-25.6, 9.7-20.5, 2.13-25.2, 1.5-14.1 m (Jerves-Cobo et al., 2020;Jerves-Cobo et al., 2018b). (line 77-79)

**9. L120**: Using the headspace equilibrium method to measure pCO2 with no consideration of al-kalinity/DIC concentration is prone to errors and may lead to gross biases in the finally calculated pCO2 results. The authors can take a look on this recent open discussion

(https://bg.copernicus.org/preprints/bg-2020-307/) and, if necessary, correct your calculations after headspace equilibration.

We appreciated your recommended open discussion and thoroughly read the preprint. The correction of the headspace equilibrium method should be applied in the undersaturated samples as written in the abstract of the preprint : "the simple headspace calculations can lead to high error (up to -800%) or even impossible negative values in highly undersaturated samples equilibrated with ambient air unless the shift in carbonate equilibrium is explicitly considered". This is not the case in our study as the obtained $pCO_2$ levels were extremely supersaturated. It was written in the revised manuscript as follows.

- Also noteworthy is that $pCO_2$ in the Cuenca urban river system were extremely supersaturated compared to the average $pCO_2$ of 3.2 matm among 47 large rivers with a near-global distribution (Cole and Caraco, 2001) or the average $pCO_2$ of 6,415 µatm from African inland waters (Borges et al., 2015). The extremely high concentration was induced by $CO_2$ supersaturation found in the sites in the Tomebamba tributary, such as sites 15, 16, and 35 with the $pCO_2$ level of 152.4, 107.2, and 60.3 matm, respectively. These were contaminated sites that had a high concentration of $NH_4^+$ of more than 15 mg $L^{-1}$ and a thick anaerobic sludge layer of 5-20 cm, causing low DO concentration of 2.7-3.4 mg $L^{-1}$. The contamination was induced by the discharges from polluted brooks, i.e. Milchichig Brook, El Valle Brook, and Saucay Brook, into which domestic wastewater has been discharged directly without any treatment (Jerves-Cobo et al., 2020). Several sites with extremely high dissolved concentrations of GHG were also found in the Tarqui tributary. For instance, the $pCO_2$ level reached 86.9 matm in site 02 in the Tarqui tributary where the highest concentrations of TN, $NH_4$, turbidity, and TDS, together with low DO level of 5.54 mg $L^{-1}$ were found. Also supersaturated by $CO_2$, site 22 was another polluted site in Tarqui tributary that was also characterized by very stagnant water with flow velocity of 0.01 m $s^{-1}$ and low DO level of 2.7 mg $L^{-1}$. (line 201-214)

**10. L157**: To calculate the total emissions of each river tributary, shouldn't it be based on the total stream surface area? I don't know how the total emission flux was calculated on the basis of total watershed area. If it is the total stream surface area used in the calculations, the authors need to elaborate on the details on the stream surface area estimation.

Indeed, the calculation of the total emissions of each river tributary was based on the total stream surface area of each tributary. However, we removed the estimation of annual emissions from the rivers as our sampling campaign was unable to cover seasonal variation of the emissions which can lead to high uncertainty in our calculation.

**11. L165**: Holgerson and Raymond (2016) only looked at lakes (reservoirs), not at rivers and streams. Thus the results here are not directly comparable to Holgerson and Raymond (2016).

We removed this reference.

**12. L171**: the authors need to justify why these two WQIs were used in this study. A brief justification will suffice.

We added the justification of the application of the two WQIs in our study as follows.

- By aggregating the measurements of multiple water quality parameters, WQI as a single number can be used to assess the quality of a water resource for serving different purposes (Lumb et al., 2011). However, as each WQI has its own demerits, no WQIs can be universally applicable (Tyagi et al., 2013). Among 30 WQIs listed in Sutadian et al. (2016), Prati and Oregon indexes were chosen in this study because of their successful establishment for river water quality assessment (Zotou et al., 2019) and their required parameters were measured during the sampling campaigns. Other WQIs, such as National Sanitation Foundation (NSF) WQI or Stoner's index, need parameters that were not measured in this study. Additionally, despite their suitable required parameters, Canadian Council of Ministers of Environment (CCME) index and Weighted Arithmetic WQI were not applied due to their common application in drinking water use (Sutadian et al., 2016) and their limitations listed in Tyagi et al. (2013). (line 126-134)

**13. L222-225**: how were these relative contributions calculated? Because the five tributary catchments have different catchment sizes, it is reasonable to observe a large contribution from the Tomebamba tributary because of its large catchment size. I don't think such comparison makes sense as it is not normalized by catchment area. In need, because almost all the sampling sites are located downstream near the catchment outlet, the calculated GHG emission fluxes are site-specific and don't really reflect the spatial variability of GHG emissions across the five tributaries or across the whole study area.

We calculated the contribution of each tributary as follows. First, we estimated the mean value of the GHG emissions (in $CO_2$ equivalent) from each tributary by multiplying the mean value of the fluxes from each tributary with its watershed area. Subsequently, we calculate the contribution of each tributary by dividing this value by the sum of the GHG emissions from each tributary. Indeed, by this calculation, the contribution of each tributary took into account its catchment area and we did not compare the fluxes from the tributaries but compare the amount of GHGs emitted from each tributary. From the obtained results, the contribution of Tomebamba tributary was the highest although its area ($113.03$ km$^2$) is not the largest compared to the area of Tarqui tributary ($138.98$ km$^2$),

However, as mentioned in the previous responses, since the sampling campaign could not cover the seasonal variation of the emissions from the river basin, we removed this calculation. This also abridged the manuscript and keep its story more focused and concise.

**14. L225-227**: following my comment above, how was the spatial variation determined? By comparing the sites within each tributary catchment? This spatial variation is reasonably clear as the sites are located within different land use landscapes or with/without wastewater inputs.

As explained above, we compared the contribution of each tributary to investigate how much the contribution varied from one tributary to the others. Each tributary has a specific type of land use that can affect their GHG emissions. For example, Machangara and Yanuncay tributaries have mainly natural areas while Tomebamba tributary runs through the city of Cuenca and Cuenca tributary is a discharge of a wastewater treatment plant that purifies domestic wastewater of the whole city. From the comparison, we were able to gain more insights into the spatial variation of GHG emissions as a function of land-use types.

**15. L230**: The skewness was clearly caused the extremely high values. I am not sure if you should remove these outliers for this plotting. If the outliers are removed, the arithmetic means will be lower

Indeed, the extremely high values led to the skewness of the data distribution and higher value of the arithmetic means. However, we intentionally showed the values to indicate the much higher dissolved gas concentrations in the sites where we specifically analyzed the reason for the elevation. In fact, these outliers were very informative as they represent the sites with very poor water quality indicated by high organic matter and nutrients concentrations, stagnant water, and low oxygen concentrations. As such, the results highlighted the effects of water quality of the sites and their land covers on their dissolved gas concentrations. We described this in detail in the revised manuscript as follows.

- The extremely high concentration was induced by $CO_2$ supersaturation found in the sites in the Tomebamba tributary, such as sites 15, 16, and 35 with the $pCO_2$ level of 152.4, 107.2, and 60.3 matm, respectively. These were contaminated sites that had a high concentration of $NH_4^+$ of more than 15 mg $L^{-1}$ and a thick anaerobic sludge layer of 5-20 cm, causing low DO concentration of 2.7-3.4 mg $L^{-1}$. The contamination was induced by the discharges from polluted brooks, i.e. Milchichig Brook, El Valle Brook, and Saucay Brook, into which domestic wastewater has been discharged directly without any treatment (Jerves-Cobo et al., 2020). Several sites with extremely high dissolved concentrations of GHG were also found in the Tarqui tributary. For instance, the $pCO_2$ level reached 86.9 matm in site 02 in the Tarqui tributary where the highest concentrations of TN, $NH_4$, turbidity, and TDS, together with low DO level of 5.54 mg $L^{-1}$ were found. Also supersaturated by $CO_2$, site 22 was another polluted site in Tarqui tributary that was also characterized by very stagnant water with flow velocity of 0.01 m $s^{-1}$ and low DO level of 2.7 mg $L^{-1}$. Note that the mean value of the samples collected from Tomebamba, Tarqui and Cuenca were much higher than the median value, indicating the dissolved GHG concentrations in the tributaries were positively skewed. The skewness was caused by several extremely high dissolved GHG concentrations in the abovementioned sites located in the three tributaries. (line 204-216)

On the other hand, we log10 transformed and standardized the obtained dissolved GHG concentrations prior to the application of the linear mixed models and random forests. Moreover, the normality and residual diagnostic were also performed to check the assumptions of the statistical analyses, as written in the revised manuscript as follows.

- The dissolved GHG concentrations were log10 transformed and standardized. A final check for normality was done by using Cleveland plots (Supplementary Material S4). Moreover, homogeneity was checked via the residuals of the fitted model (Supplementary Material S5) while the assumption of multicollinearity was omitted due to the absence of fixed parameters. (line 164-168)

**16. L263**: this is a very good observation. Does the dissolved GHG concentration also show a similar variation by water quality level? Also, for the DO, does it also show a similar variation? Because GHG emission reflects the combined effect of dissolved GHG concentration and gas transfer velocity, it will

be more meaningful to show the change of dissolved GHG concentrations which are perhaps better indicative of the water quality status.

The dissolved GHG concentrations showed the same trend and similar variations over different water quality levels and different land-use types in the revised manuscript in which we removed the flux calculation and used the dissolved GHG concentrations instead. Here are new Figures showing the dissolved GHG concentrations in different water quality levels and land-use types in the revised manuscript.

[Figure]

Figure 3. Dissolved concentrations of the three greenhouse gases from the Cuenca urban river system in different water quality categories using Oregon and Prati Indexes. Box plots display 10th, 25th, 50th, 75th and 90th percentiles, and individual data points outside the 10th and 90th percentiles. Blue dots represent the mean of the dissolved concentrations in the water quality categories.

[Figure]

Figure 4. Dissolved concentrations of the three greenhouse gases from the Cuenca urban river system in different land-use types. Box plots display 10th, 25th, 50th, 75th and 90th percentiles, and individual data points outside the 10th and 90th percentiles. Blue dots represent the arithmetic mean of the dissolved concentrations from different land use categories.

Regarding DO, this is a very good suggestion to further analyze the role of DO in the production of GHGs in the rivers. Since DO plays an essential role in the calculation of the two WQI indexes, we obtained a similar trend of DO concentration over different water quality levels in which the more polluted the streams, the lower the DO values (below figure). The variation also appeared more clearly in case of Prati Index. We added this discussion in the revised manuscript and the figure in the revised supplementary material as follows.

- Since DO plays an essential role in the calculation of the two WQI indexes, we obtained a similar trend of DO concentration over different water quality levels in which the more polluted the streams, the lower the DO values (Figure S7.1). (line 255-257)

[Figure]

Figure S7.1. Dissolved oxygen concentrations in different water quality categories using Oregon and Prati Indexes. Box plots display 10th, 25th, 50th, 75th and 90th percentiles, and individual data points outside the 10th and 90th percentiles. Blue dots represent the mean of the concentrations in different water quality categories. Note that y axis is inverted.

**17. L274**: again, the cited values from Holgerson and Raymond, 2016 are for lakes only, not for rivers.

We removed this reference.

**18. L275**: To estimate the global average GWP, perhaps you need to know the relative abundance of streams/rivers/lakes/reservoirs/wetlands.

We removed this estimation due to its high uncertainty and numerous assumptions.

**19. L291-303**: Move these descriptions to the section 'study area'?

We removed this to new section **2.5 Land use and land cover** in the Materials and Methods to briefly introduce the land-use types of the Cuenca river basin and the method in which we evaluate the effects of different land-use types on the dissolved GHG concentrations as follows.

- Due to the large sampling area, land-use types widely varied while other hydro-morphological variables remained relatively stable across the five tributaries. Particularly, urban and resident areas were dominant with around 55% of the total sampling areas, while forest and agriculture occupied 8-11% and 14-20%, respectively. Minor sampling area was surrounded by industrial factories and construction sites, with less than 5% each. Several riversides were next to the road, occupying 11-19% of the total sampling area. The distribution of the land-use types was not evenly among the rivers. Intensive urban activities can be found near the Cuenca and from the middle to the end of Tomebamba rivers. Conversely, Yanuncay and Machangara cross two natural reserves, i.e. Cajas National Park and the Machangara-Tomebamba protected forest, leading to their pristine water quality conditions. In addition, these two aforementioned rivers cross the city of Cuenca in the latest part of their path before the confluence with the Tomebamba River (Jerves-Cobo et al., 2018b). In addition, these two aforementioned rivers cross the city of Cuenca in the latest part of their path before the confluence with the Tomebamba River. Tarqui river locates near agricultural irrigation and livestock production areas, causing their high nutrient and organic inputs (Jerves-Cobo et al., 2018b). To evaluate the effects of land use on the dissolved GHG concentrations of the tributaries, we considered five different types of land use, i.e. nature (close to the forests), industry (close to factories, and mining areas), agriculture (close to arable land, orchard, and farms), roads, and urban areas (close to residential and urban areas). (line 145-156)

**20. L308**: for the effect of land-use types on GHG emissions, I agree with the authors that land use will have diverse impacts on GHG concentration and emissions. However, most of the sampling sites (Fig 1) are nested within the catchment. This suggests that the observed GHG emissions at these sites are not necessarily affected or controlled by only one single land use type. For example, the sites in the downstream may have been simultaneously affected by urban, agriculture and nature. If without an accurate quantification of their relative contributions, it is problematic to compare the impacts of different land use types on GHG emissions as shown in Fig 4. This is also related to the results in Table 2. Are these GWPs solely governed by one land use type?

We agree that the effects of water quality and land use on GHG production of a site can be affected by these factors from the previous sites in the same river. It is normally referred to as spatial autocorrelation

that appears when the values of data sampled at the same location exhibit more similar patterns than those further apart. As such, we applied a linear mixed model to investigate the spatial variation of the dissolved GHG concentrations. The obtained results showed that high spatial autocorrelation of the dissolved concentrations within a tributary, which verifies your concern. However, as stated in the revised manuscript, the distribution of the land-use types was not evenly among the rivers. Particularly, intensive urban activities can be found near the Cuenca and from the middle to the end of Tomebamba rivers. Conversely, Yanuncay and Machangara cross two natural reserves, i.e. Cajas National Park and the Machangara-Tomebamba protected forest, leading to their pristine water quality conditions. Tarqui river locates near agricultural irrigation and livestock production areas, causing their high nutrient and organic inputs. From this point of view, together with the results showed in Figure 4 of dissolved GHG concentrations in different land-use types, the effects of land-use types and land cover appeared obvious. The description of the land-use types was written in new section 2.5 Land use and land cover in Materials and Methods as follows.

- The distribution of the land-use types was not evenly among the rivers. Intensive urban activities can be found near the Cuenca and from the middle to the end of Tomebamba rivers. Conversely, Yanuncay and Machangara cross two natural reserves, i.e. Cajas National Park and the Machangara-Tomebamba protected forest, leading to their pristine water quality conditions. In addition, these two aforementioned rivers cross the city of Cuenca in the latest part of their path before the confluence with the Tomebamba River (Jerves-Cobo et al., 2018b). In addition, these two aforementioned rivers cross the city of Cuenca in the latest part of their path before the confluence with the Tomebamba River. Tarqui river locates near agricultural irrigation and livestock production areas, causing their high nutrient and organic inputs (Jerves-Cobo et al., 2018b). (line 149-154)

Moreover, as showing in the outliers mentioned in the responses of the 15[th] specific comment, the dissolved GHG concentrations of the sites were locally affected by their water quality. For example, the sites with high dissolved GHG concentrations, such as 15, 16, and 35 in Tomebamba, were very contaminated due to the discharges of the polluted Milchichig Brook, El Valle Brook, and Saucay Brook, respectively, where domestic wastewater has directly been discharged into without any treatment. The discharges led to high concentration of $NH_4^+$ of more than 15 mg $L^{-1}$ and a thick anaerobic sludge layer of 5-20 cm, causing low DO concentration of 2.7-3.4 mg $L^{-1}$ in the sites. On the other hand, due to large study area and self-purification capacity of the river, the water quality of sites 12, 34, and 13, which are close to sites 15, 16 and 35, was acceptable according to Prati Index.

**21. L368-370**: how have these factors affected the variation of the GHG emissions? Any evidence?

The important analysis of the random forests showed these hydraulic variables among the most important variables explaining the variation of the dissolved GHG concentrations. The elevation of GHG emissions from inland waters due to the changes in river flow characteristics has been illustrated in numerous studies investigating the GHG emissions from hydropower dams. We deliberated the potential effect of the variables on the dissolved GHG concentrations in the revised manuscript as follows.

- It was also showed in the important analysis that other factors indicating flow characteristics, such as turbidity, average velocity, average depth, and water temperature were among the most important variable explaining the variation of the dissolved GHG concentrations from the rivers. Particularly, the turbulence of the river flow appeared to affect the variation of $CH_4$ the most among the three gases. The elevation of GHG emissions from inland waters due to the changes in river flow characteristics have been illustrated in numerous studies investigating the GHG emissions from hydropower dams (Rasanen et al., 2018;Fearnside, 2016) (line 331-336)

**22. L390**: again, for these emission rates expressed as Gg CO2 yr-1, I don't think it is reliable as expected. The sampling was not spatially and temporal resolved enough for an annual-scale estimation.

As mentioned in the previous responses, we removed the estimation of annual emissions from the tributaries.

**23. L428**: what does the 'later' refer to? I believe it is 'nature'. Fig 1. How did the authors differentiate rivers from streams? By Strahler order?

That should have been written as "latter" which referred to nature sites. We did not differentiate rivers from streams by their width or Strahler order. We replaced the terms 'streams' with 'rivers' in line 315 to avoid potential confusion from the readers.

**References**

Borges, A. V., Darchambeau, F., Teodoru, C. R., Marwick, T. R., Tamooh, F., Geeraert, N., Omengo, F. O., Guerin, F., Lambert, T., Morana, C., Okuku, E., and Bouillon, S.: Globally significant greenhouse-gas emissions from African inland waters, Nat Geosci, 8, 637-+, 10.1038/Ngeo2486, 2015.

Cole, J. J., and Caraco, N. F.: Carbon in catchments: connecting terrestrial carbon losses with aquatic metabolism, Mar Freshwater Res, 52, 101-110, Doi 10.1071/Mf00084, 2001.

Daelman, M. R., van Voorthuizen, E. M., van Dongen, L. G., Volcke, E. I., and van Loosdrecht, M. C.: Methane and nitrous oxide emissions from municipal wastewater treatment - results from a long-term study, Water Sci Technol, 67, 2350-2355, 10.2166/wst.2013.109, 2013.

Drake, T. W., Raymond, P. A., and Spencer, R. G. M.: Terrestrial carbon inputs to inland waters: A current synthesis of estimates and uncertainty, Limnol Oceanogr Lett, 3, 132-142, 10.1002/lol2.10055, 2018.

Fearnside, P. M.: Greenhouse gas emissions from Brazil's Amazonian hydroelectric dams, Environ Res Lett, 11, Artn 011002

10.1088/1748-9326/11/1/011002, 2016.

Hotchkiss, E. R., Hall Jr, R. O., Sponseller, R. A., Butman, D., Klaminder, J., Laudon, H., Rosvall, M., and Karlsson, J.: Sources of and processes controlling CO2 emissions change with the size of streams and rivers, Nat Geosci, 8, 696, 10.1038/ngeo2507, 2015.

Jerves-Cobo, R., Cordova-Vela, G., Iniguez-Vela, X., Diaz-Granda, C., Van Echelpoel, W., Cisneros, F., Nopens, I., and Goethals, P. L. M.: Model-Based Analysis of the Potential of Macroinvertebrates as Indicators for Microbial Pathogens in Rivers, Water-Sui, 10, 10.3390/W10040375, 2018a.

Jerves-Cobo, R., Lock, K., Van Butsel, J., Pauta, G., Cisneros, F., Nopens, I., and Goethals, P. L. M.: Biological impact assessment of sewage outfalls in the urbanized area of the Cuenca River basin (Ecuador) in two different seasons, Limnologica, 71, 8-28, 10.1016/j.limno.2018.05.003, 2018b.

Jerves-Cobo, R., Forio, M. A. E., Lock, K., Van Butsel, J., Pauta, G., Cisneros, F., Nopens, I., and Goethals, P. L. M.: Biological water quality in tropical rivers during dry and rainy seasons: A model-based analysis, Ecological Indicators, 108, UNSP 105769

10.1016/j.ecolind.2019.105769, 2020.

Lumb, A., Sharma, T. C., and Bibeault, J. F.: A Review of Genesis and Evolution of Water Quality Index (WQI) and Some Future Directions, Water Qual Expos Hea, 3, 11-24, 10.1007/s12403-011-0040-0, 2011.

Musenze, R. S., Werner, U., Grinham, A., Udy, J., and Yuan, Z. G.: Methane and nitrous oxide emissions from a subtropical estuary (the Brisbane River estuary, Australia), Sci Total Environ, 472, 719-729, 10.1016/j.scitotenv.2013.11.085, 2014.

Qu, B., Aho, K. S., Li, C. L., Kang, S. C., Sillanpaa, M., Yan, F. P., and Raymond, P. A.: Greenhouse gases emissions in rivers of the Tibetan Plateau, Sci Rep-Uk, 7, 10.1038/S41598-017-16552-6, 2017.

Rasanen, T. A., Varis, O., Scherer, L., and Kummu, M.: Greenhouse gas emissions of hydropower in the Mekong River Basin, Environ Res Lett, 13, Artn 034030

10.1088/1748-9326/Aaa817, 2018.

Raymond, P. A., Zappa, C. J., Butman, D., Bott, T. L., Potter, J., Mulholland, P., Laursen, A. E., McDowell, W. H., and Newbold, D.: Scaling the gas transfer velocity and hydraulic geometry in streams and small rivers, Limnology and Oceanography: Fluids and Environments, 2, 41-53, 10.1215/21573689-1597669, 2012.

Sutadian, A. D., Muttil, N., Yilmaz, A. G., and Perera, B. J. C.: Development of river water quality indices-a review, Environ Monit Assess, 188, 10.1007/s10661-015-5050-0, 2016.

Tyagi, S., Sharma, B., Singh, P., and Dobhal, R.: Water Quality Assessment in Terms of Water Quality Index, American Journal of Water Resources, 1, 34-38, 2013.

Zotou, I., Tsihrintzis, V. A., and Gikas, G. D.: Performance of Seven Water Quality Indices (WQIs) in a Mediterranean River, Environ Monit Assess, 191, 10.1007/s10661-019-7652-4, 2019.